# GEMS v1.0: Generalizable empirical model of snow accumulation and melt based on daily snow mass changes in response to climate and topographic drivers

Atabek Umirbekov[1,2,5], Richard Essery[3], Daniel Müller[1,2,4]

[1] Leibniz Institute of Agricultural Development in Transition Economies (IAMO), Theodor-Lieser-Str. 2, 06120 Halle (Saale), Germany
[2] Geography Department, Humboldt-Universität zu Berlin, Unter den Linden 6, 10099 Berlin, Germany
[3] School of Geosciences, University of Edinburgh, EH9 3JW, Edinburgh, United Kingdom
[4] Integrative Research Institute on Transformations of Human-Environment Systems (IRI THESys), Humboldt Universität-zu-Berlin, Berlin, Germany
[5] Tashkent Institute of Irrigation and Agricultural Mechanization Engineers (TIIAME), 39 Kari Niyazov Str., Tashkent, 100000, Uzbekistan

*Correspondence to*: Atabek Umirbekov (umirbekov@iamo.de)

**Abstract:** Snow modeling is often hampered by the availability of input and calibration data, which can affect the choice of models, their complexity, and transferability. To address the trade-off between model parsimony and transferability, we present the Generalizable Empirical Model of Snow Accumulation and Melt (GEMS), a machine learning-based model, which requires only daily precipitation, temperature or its daily diurnal cycle, and basic topographic features, to simulate snow water equivalent. The model embeds a Support Vector Regression pretrained on a large dataset of daily observations from a diverse set of the Snowpack Telemetry Network (SNOTEL) stations in the United States. GEMS does not require any user calibration, except for the option to adjust the temperature threshold for rain-snow partitioning, though the model achieves robust simulation results with the default value. We validated the model with long term daily observations from numerous independent SNOTEL stations not included in the training and with data from reference stations of the Earth System Model-Snow Model Intercomparison Project. We demonstrate how the model advances large scale SWE modelling in regions with complex terrain that lack in-situ snow mass observations for calibration, such as the Pamir and Andes, by assessing the model`s ability to reproduce daily snow cover dynamics. Future model improvements should consider the effects of vegetation, improve simulation accuracy for shallow snow in warm locations at lower elevations and possibly address wind-induced snow redistribution. Overall, GEMS provides a new approach for snow modeling that can be useful for hydro-climatic research and operational monitoring in regions where in-situ snow observations are scarce.

## 1. Introduction

Snow is a vital component of the global climate system and plays a key role in regulating the temperature of the Earth's surface and in governing the hydrologic cycle on both global and regional scales (Zhang, 2005; Sturm et al., 2017). Furthermore, snow plays an important role as a natural means of water storage and supply for human activities (Barnett et al., 2005), with a substantial share of the world's population relying on snowmelt to provide water for agriculture and domestic needs (Mankin et al., 2015; Kraaijenbrink et al., 2021). Snowmelt is particularly crucial for densely populated downstream areas, where the timing and quantity of snow accumulation and melting in mountainous regions determine the availability of water (Armstrong et al., 2019; Immerzeel et al., 2020). Accurate estimation of snow mass accumulation and melt is therefore essential for water resource management as well as for early warning of droughts and floods (Beniston, 2008).

Energy-balance and temperature-index snow models are the two main types of models to simulate snow accumulation and melting. Energy-balance snow models, also referred to as physics-based models, calculate the amount of snow mass based on the balance between the energy input to the snowpack and the energy output from the snowpack (Essery, 2019). These models consider multiple factors such as incoming solar radiation, air temperature, humidity, precipitation, and wind speed, as well as the physical properties of the snowpack, such as snow density and surface albedo. Due to high input data requirement of energy-balance models, which are often lacking especially in countries of the Global South, researchers often opt for relatively simpler conceptual temperature-index models, which rely on temperature and precipitation data (Hock, 2003; Ohmura, 2001). These models estimate the amount of snowmelt by determining empirical relationship between temperature and amount of snowmelt (Link et al., 2019). The two types of snow models usually require adjustment of internal parameters that characterize embedded snow processes. Depending on the complexity of a model, calibrating its parameters can often become a computational burden and introduces challenge of model parameters equifinality (Beven, 1993, 2006; Günther et al., 2020).

Despite the differences in the number of internal processes represented and the corresponding data requirements, both types of models produce similar results when calibrated and applied to the same spatial domain and same climatic conditions (Kumar et al., 2013; Bavera et al., 2014; Magnusson et al., 2011; Shakoor et al., 2018). The growing number of the intercomparison studies conclude that model complexity does not determine performance (Essery et al., 2013; Magnusson et al., 2015; Menard et al., 2021), and simpler models may perform equally well or even outperform more sophisticated snow models in some cases, e.g. when input data is of low quality (Terzago et al., 2020). Models calibrated to the same climate conditions can however produce different simulations under different climate conditions (Carletti et al., 2022). In this regard, physics-based snow models are known to show better temporal and spatial transferability than temperature-index models (Magnusson et al., 2015), since they are able to capture the dynamic physical processes that govern formation, accumulation, and melting of snow, which allows them to simulate snow under a wide range of climate conditions. The generalizability and transferability of snow models are important considerations in their development and deployment, especially for applications over geographical domains where in-situ snow-measurements are non-existent or scarce.

In recent years, the research community saw an emergence of so-called data-driven approaches for snow modeling, which usually employ machine learning techniques on extensive sets of snow observations and predictor variables. In terms of ways in which machine learning (ML) has been applied for snowpack modeling, the respective research studies can be grouped into several main approaches. One common approach is estimating the spatial distribution of snowpack by applying ML-supported *interpolation* of sparse snow observations and using topographical features, meteorological and satellite data (Broxton et al., 2019; Mital et al., 2022). Other studies have explored the potential of satellite radar data for direct *detection* of instantaneous properties of snowpack (Santi et al., 2022; Daudt et al., 2023). In cases where several gridded snow products are available, ML can be employed for a better prediction through *assimilation* of multiple estimates or *bias-correction* (Shao et al., 2022; King et al., 2020). A few recent studies applied ML in a manner consistent with traditional snow models, explicitly modeling snow mass accumulation and melt dynamics (Vafakhah et al., 2022; Duan et al., 2023; Wang et al., 2022). However, most of the noted approaches also rely on is-situ observations or extensive set of regional reanalysis variables, which restricts their wider applicability due to unavailability of such data in many regions. Furthermore, the ability of pretrained machine learning models to generalize to new geographic and climatic domains remains another challenge; machine learning models often perform less well outside the data distribution used to train them (Chase et al., 2022; Hernanz et al., 2022).

We address these challenges with the Generalizable Empirical Model of Snow accumulation and melt (GEMS) that, by leveraging the power of machine learning to learn from a large number of diverse experiments, generates accurate estimates of snow water equivalent from a limited range of input data. Instead of modeling snow as a dynamic system, the GEMS employs assimilated statistical relationship between changes in snow mass in response to climate variables while accounting for topographic features. By incorporating diverse climate and topographic observations into the model training, we demonstrate how it simulates snow water equivalent with acceptable accuracy even in distant out-of-sample geographical locations.

## 2.  Model description

### 2.1  Model structure and required inputs

GEMS is an empirical model based on statistical learning of daily changes in snow water equivalent in response to precipitation, temperature, and topography. It incorporates Support Vector Regression (SVR) that was trained using more than 28,000 observations of daily snow accumulation and melt from 94 stations of the Snowpack Telemetry Network (SNOTEL) in the United States. The model has only one adjustable parameter, a temperature threshold ($T_S$) that specifies when 100% of precipitation falls as snow, which is used to confine the SVR simulations during the rain-to-snow transition and snow accumulation phases. **Figure *1*** depicts the model's workflow and its primary components, which are described in greater detail in the following sections.

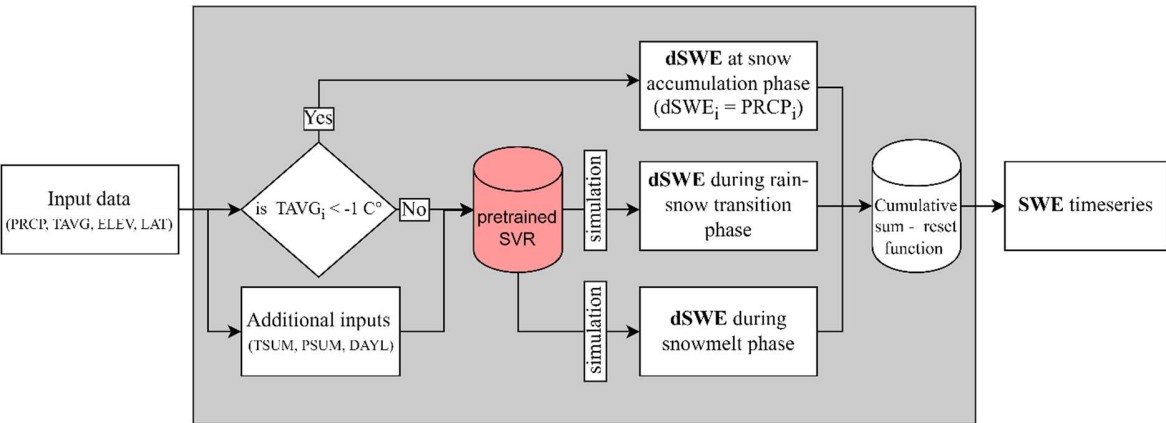

**Figure 1.** GEMS workflow. Model elements and abbreviations are described in the sub-sections that follow

The GEMS v1.0 model is developed in the R programming environment (R Core Team, 2020), with anticipated replication in Python and possibly other program languages. It is available as a pretrained SVR model, accompanied by an R script containing a set of functions that take input data on daily time steps, calculate additional predictors, and generate corresponding estimates of snow water equivalent. It can be applied for both single-point and spatially

distributed simulations by feeding input data in tabular form or raster files, respectively.

The model is available in four variations of the required input data listed in **Table 1**. The simplest one, GEMS-4P (the "P" suffix specifies the number of required inputs), requires four predictors, such as daily precipitation, average temperature, latitude, and elevation. Three other modifications, GEMS-5P, GEMS-6P and GEMS-7P, require additional predictors, such as daily diurnal temperature range (daily maximum and minimum temperatures) and a

location-specific heat-insolation index, which can be retrieved through the Google Earth Engine.

**Table 1.** Required forcing data for GEMS

| Input data | GEMS-4P | GEMS-5P | GEMS-6P | GEMS-7P |
|---|---|---|---|---|
| Precipitation (mm) | ✓ | ✓ | ✓ | ✓ |
| Mean daily temperature (°C) | ✓ | ✓ | ✓ | ✓ |
| Maximum daily temperature (°C) | | | ✓ | ✓ |
| Minimum daily temperature (°C) | | | ✓ | ✓ |
| Latitude (decimal degrees) | ✓ | ✓ | ✓ | ✓ |
| Elevation (meters a.s.l.) | ✓ | ✓ | ✓ | ✓ |
| Heat-insolation index | | ✓ | | ✓ |

## 2.2  Support vector regression

In its core embedding, GEMS is built on a pretrained SVR that estimates daily accumulation and melt of SWE given the meteorological conditions and terrain features. SVR is a supervised machine learning algorithm that projects data into a higher dimensional space, then minimizes error by generating a set of hyperplanes that explain as many observations as possible (Awad and Khanna, 2015; Vapnik and N., 1995). SVR utilizes radial basis function kernels (Schölkopf et al., 2004) and is calibrated for optimal cost and gamma hyperparameters, which govern training errors and degree of influence of a single training point. The SVR can be expressed as:

$$SVR(x) = \sum_{i=1}^{N}(\alpha_i - \alpha_i^*)\, K(x_i, x) + b \qquad (1)$$

where,

$N$ is the total number of support vectors, which corresponds to number of data points during training,

$\alpha_i, \alpha_i^*$ are Langrage multipliers, such that $\alpha_i \geq 0\ and\ \alpha_i^* \leq 0$,

$K$ is the radial basis function kernel, such that:

$$K(x_i, x_j) = \exp\left[-\frac{||x_i - x_j||^2}{2\sigma^2}\right]$$

where,

$||x_i - x_j||$ is the Euclidian distance between feature vectors corresponding to the i-th and j-th input data points.

We trained the model using data from selected SNOTEL stations (described in Section **3.1**) for 2017 and 2018, We fine-tuned the hyperparameters so that the model produces similar levels of accuracy when applied to observations from the same stations for 2019 and 2020. The hyperparameter calibration process involved an exhaustive 'grid-search' technique, which systematically explored all possible combinations within predefined parameter ranges. Ultimately, we selected the hyperparameter configurations that resulted in the lowest root mean squared error between simulated and observed dSWE during both model training on observations from 2017 and 2018 and we tested the model on observations from 2019 and 2020.

### 2.3 Temperature threshold constraint and model-wrapper function

Due to instabilities of daily changes in SWE (dSWE) estimated by the SVR during rain-snow transition phases (described in the Section **4.1**), simulated dSWE at any day (t) values are constrained as follows:

$$dSWE_t = \begin{cases} SVR(x_t), & \text{if } TAVG_t \geq T_S \\ PRCP_t, & \text{if } TAVG_t < T_S \end{cases} \qquad (2)$$

where,

$T_S$ is a 100% rain-snow temperature threshold, with default value of -1°C

The dSWE estimates are then aggregated into daily SWE timeseries using the cumulative sum-reset function:

$$SWE_t = \begin{cases} 0, & \text{if } t = 0 \\ \max(dSWE_t + SWE_{t-1}, 0), & \text{if } t > 0 \end{cases} \quad (3)$$

## 3. Data

### 3.1 Data for training support vector regression

For training the SVR, we used the SNOTEL data listed in **Table 2**, the largest network of automated weather stations that collect data on snow water equivalent, precipitation, temperature, and other climatic variables. We used daily observations from 94 SNOTEL stations located in the contiguous United States for two hydrological years, 2017 and 2018. **Figure 2** displays location of the selected stations, along with density distribution of their main geographical and topographical characteristics.

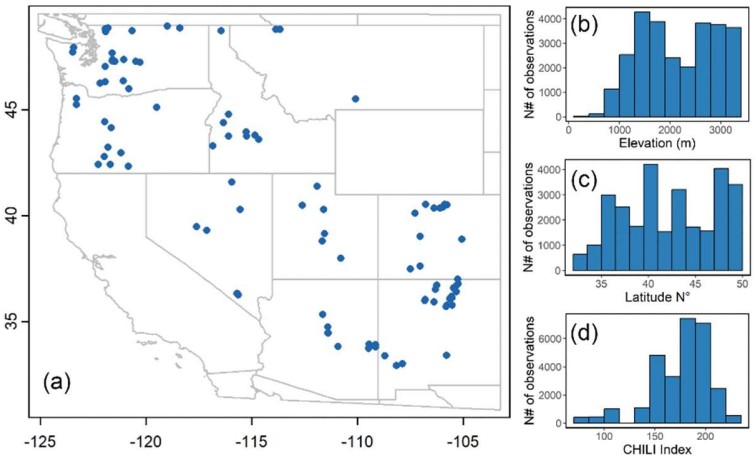

**Figure 2.** Location of SNOTEL stations used for training the SVR (a), and their density distributions in terms of (b) elevation, (c) latitude, and (d) heat-insolation index.

In the 1990s, the temperature observations from SNOTEL showed anomalous trends (Pepin et al., 2005), which were eventually attributed to a new temperature sensor (Oyler et al., 2015), installed with an incorrect equation algorithm. To correct for this bias, we applied a debiasing equation on SNOTEL temperature data proposed by Brown *et al.* (2019) and using metadata of affected stations (Air Temperature Bias Correction).

SNOTEL precipitation gauges may also be susceptible to solid precipitation undercatch, especially when snowfall occurs in windy conditions (USDA, 2014). Scalzitti et al., 2016 provide a comprehensive review of the issues associated with precipitation undercatch, highlighting reported undercatch ranging from 11% for snowfall under 2m/sec wind speed to more than 30% during intense snowstorm events. To ensure data accuracy, we cleaned the

training dataset by removing observations with inconsistencies between daily precipitation and snow mass accumulation. These inconsistencies refer to cases when the daily increase in SWE exceeded the reported daily precipitation.

The input data includes a heat-insolation index to account for the influence of topographic shading, which may result in a significant variability of surface energy balance and therefore in snowmelt rate, particularly in complex terrain. We used the Continuous Heat-Insolation Load Index (CHILI), which approximates effects of insolation and topographic shading on evapotranspiration and is determined by estimating insolation in the early afternoon at equinox sun height (Theobald et al., 2015). The Google Earth Engine provides access to CHILI data on a global scale with a horizontal resolution of 90 m. Since CHILI is a location-specific static characteristic, we also augmented the forcing data with daylength, which is a time-varying variable estimated using latitude of a location and day of a year.

**Table 2.** Climate and topographic data used to train the model

| Variable | Abbreviation | Source/reference |
|---|---|---|
| Daily change of Snow water equivalent (mm) | dSWE | SNOTEL |
| Precipitation (mm) | PRCP | SNOTEL |
| Mean daily temperature (°C) | TAVG | SNOTEL |
| Maximum daily temperature (°C) | TMAX | SNOTEL |
| Minimum daily temperature (°C) | TMIN | SNOTEL |
| Rolling sum of temperature over preceding three days (°C) | TSUM | Calculated using TAVG |
| Cumulative sum of precipitation over preceding three days (mm) | PSUM | Calculated using PRCP |
| Daylength (hours) | DAYL | Calculated as a function of latitude and day of a year (Forsythe et al., 1995) |
| Elevation (meters a.s.l.) | ELEV | SNOTEL |
| Heat-insolation index | CHILI | Global Continuous Heat-Insolation Load Index (Theobald et al., 2015) |

## 3.2 Data and procedure for evaluation of the model

The evaluation of the model performance followed a three-tiered structure.

First, we assessed the model performance using observations from SNOTEL stations that were not included in the training. The selection of stations for validation followed two main criteria: First. we excluded stations that exhibit precipitation undercatch, which we formulate as when SWE accumulated by March is greater than the accumulated precipitation during October to March. This approach enabled us to include more stations in the evaluation dataset while excluding only those hydrological years that exhibited inconsistencies between these variables. We selected evaluation observations using this criterion without any specific threshold for the magnitude of inconsistencies, nor did we make corrections to the precipitation time series. Out of the filtered stations we selected only stations that have complete daily observations for at least five water years, defined as October of the preceding year to September next year for any year from 2011 to 2022.  The selection algorithm filtered 520 stations from a total of approximately 703 contiguous US SNOTEL stations that had not been used for model training.

Second, we evaluated the model performance using snow and meteorological data from seven reference stations, which were used in the Earth System Model-Snow Model Intercomparison Project (ESM-SnowMIP), hereinafter referred to as ESM-SnowMIP reference stations. **Table 3** below provides descriptions of these sites.

**Table 3.** Geographic and climate characteristics of the ESM-SnowMIP reference stations

| Site name, country | Abbreviation | Latitude (°N) | Elevation (meters a.s.l.) | Snow cover classification | Köppen climate classification |
|---|---|---|---|---|---|
| Col de Porte, France | CDP | 45.3 | 1,325 | Alpine | Warm-summer humid continental climate |
| Old Aspen, Canada | OAS | 53.63 | 600 | Taiga | Warm-summer humid continental climate |
| Old Black Spruce, Canada | OBS | 53.99 | 629 | Taiga | Warm-summer humid continental climate |
| Old Jack Pine, Canada | OJP | 53.92 | 579 | Taiga | Warm-summer humid continental climate |
| Reynolds Mountain, USA | RME | 43.19 | 2,060 | Alpine | Warm-summer humid continental climate |
| Sapporo, Japan | SAP | 43.08 | 15 | Maritime | Hot summer continental climates |
| Senator Bec, USA | SNB | 37.91 | 3,714 | Alpine | Polar and alpine (montane) climates |
| Swamp Angel, USA | SWA | 37.91 | 3,371 | Alpine | Subarctic climate |
| Sodankylä, Finland | SOD | 67.37 | 179 | Taiga | Subarctic climate |
| Weissfluhjoch, Switzerland | WFJ | 46.83 | 2,536 | Alpine | Polar and alpine (montane) climates |

Finally, we assessed the performance of the model using distributed large-scale climate data over Western Pamir in Central Asia and Central Andes regions with complex terrain (**Figure 3**) by comparing observed and simulated snow cover. Both selected regions are characterized by semi-arid climate conditions in higher elevations and predominantly arid climate conditions in plains. We used temperature and precipitation data at 1 km resolution from CHELSA-W5E5 dataset (Karger et al., 2022a) to force the model, and compared the extent of SWE simulated during the two consecutive snow seasons between 2014 and 2016 with MODIS-derived snow cover retrievals using the cloud-gap filled MOD10A1F product images (Riggs et al., 2019).


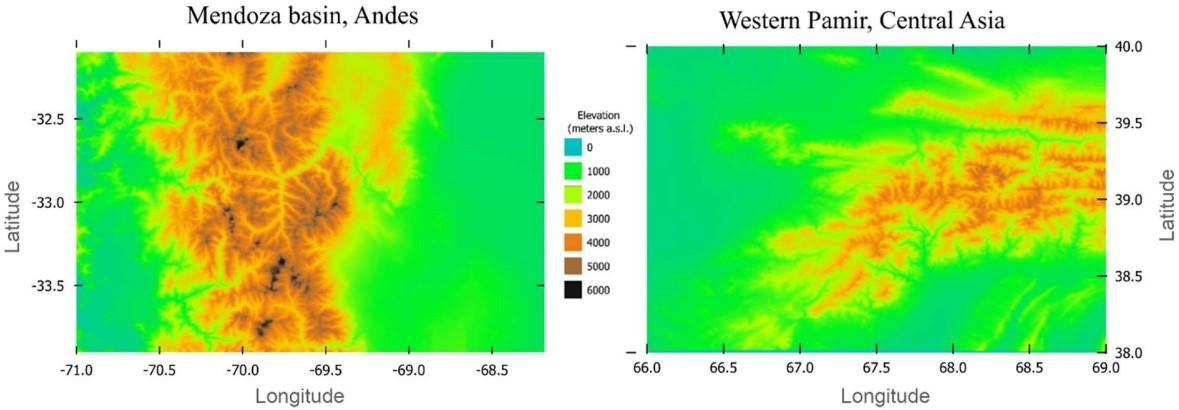

**Figure 3.** Selected regions for distributed snow modelling

The evaluation metrics for single point simulations across SNOTEL and ESM-SnowMIP reference sites consist of the Nash–Sutcliffe Efficiency (NSE) coefficient (Nash and Sutcliffe, 1970) , mean absolute percentage error of peak SWE (maxSWE MAPE), bias of the simulated peak SWE (maxSWE BIAS), and difference in snow melt-out dates:

$$NSE\left(SWE, \widehat{SWE}\right) = 1 - \frac{\sum_{i=1}^{N(days)} (SWE_i - \widehat{SWE_i})^2}{\sum_{i=1}^{N(days)} \left(SWE_i - mean(SWE_i)\right)^2}$$

where,

$SWE_i$ – observed daily SWE;

$\widehat{SWE_i}$ – simulated daily SWE

$$maxSWE\ MAPE(y, \hat{y}) = \frac{100\%}{N\ (years)} \sum_{w=1}^{N\ (years)} \frac{|y_w - \hat{y}_w|}{y_w}$$

$$maxSWE\ BIAS(y, \hat{y}) = \frac{100\%}{N(years)} \sum_{w=1}^{N(years)} \frac{y_w - \hat{y}_w}{y_w}$$

where,

$y_w$– observed peak SWE in w$^{th}$ hydrological year;

$\hat{y}_w$ – simulated peak SWE in w$^{th}$ hydrological year


$$Snow\ melt\ out\ date\ error = \frac{1}{N(years)} \sum_{w=1}^{N(years)} mdate_w - \widehat{mdate}_w$$

where,

$mdate_w$– actual date of snow disappearance in wth hydrological year;

$\widehat{mdate}_w$– date of the snow disappearance according to model simulations


All simulations for the evaluation are implemented with the GEMS-7P version of the model that uses seven predictors (**Table 1**). The Section 4.7 ("**Performance of GEMS model under different input requirements**") compares the overall performance of the model's four different versions (GEMS-7P, GEMS-6P, GEMS-5P, and GEMS-4P).

**4.   Model evaluation**

**4.1 Observed and modeled daily changes in SWE across training and validation SNOTEL stations**

**Figure 4** compares observed and predicted dSWE values obtained by running the pretrained SVR using training, calibration, and validation datasets. The model yields plausible estimates of the dSWE, albeit the variance is greater at higher melt rates. There is a greater variance between simulated and observed values in the validation dataset,

although it should be noted that it has a much larger number of observations compared to the training and calibration datasets (1.36 million, 28,600, and 32,600 observations, respectively), which results in more outliers. In each of the three instances, the slope of the linear regression between the observed and simulated values ranges between 1.03 to 0.99.

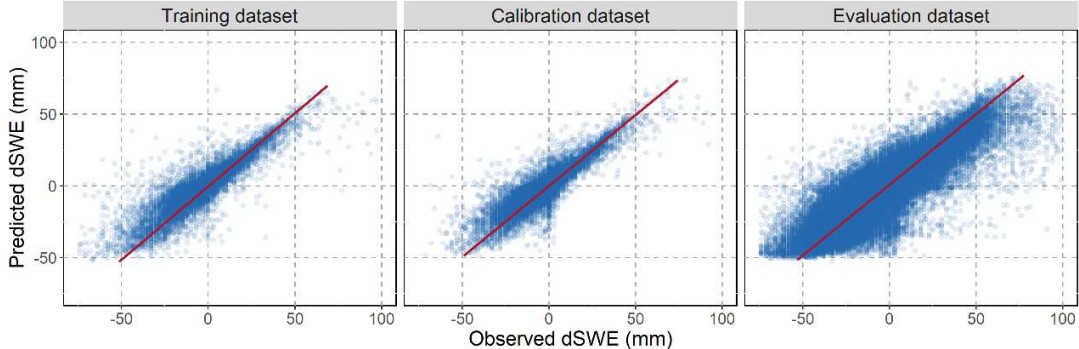

**Figure 4.** Predicted and observed dSWE values for training, calibration, and validation datasets. The red line represents the slope of the linear regression run on observed and predicted dSWE values

The validation dataset simulations exhibited a bigger proportion of outliers in the upper tile corresponding to snow accumulation phase (dSWE>0). To determine the accuracy of the SVR's performance for this phase, we compared model simulations using a sample of the validation dataset that includes observations with incremental changes of SWE at the beginning of the snow season. Since the SNOTEL observations do not contain explicit information on precipitation-snow transition, we decided to use a sample of the dataset to simulate the transition depending on climate inputs (temperature variables) and topographical characteristics (e.g. elevation). More specifically we have filtered the SNOTEL observations that closely fall on precipitation-snow transition phase by selecting observations that meet the following non-exhaustive main criteria: 1) observations for October or November when precipitation is non-zero 2) average temperature (TAVG) is less than 10 or higher than -10°C, 3) accumulated SWE is less than 20mm. We then run the model using the obtained sample of observations and estimated solid fraction of precipitation simulated by the model, i.e. amount of dSWE estimated by the model in respect to precipitation amount.

**Figure 5** depicts the rain-to-snow transition modelled using the metadata of the 520 validation SNOTEL stations. We conclude that average daily temperatures (TAVG) at which the model predicts precipitation to fall partially as snow may range from -5 to more than 5 °C and have a relatively higher association with maximum temperature and elevation. The comparison also reveals that the simulations tend to underestimate snow accumulation, since in some cases the solid component of precipitation in simulations does not reach 100% even at temperatures below -5 °C. In this regard, we have introduced a constraint (specified above in the Section **3.4** "Temperature threshold constraint and model-wrapper function"), which imposes that any daily precipitation after a certain temperature threshold ($T_S$) is considered to fall as 100% snow. We set the default value of $T_S$ as -1 °C, which the simulations revealed to be the optimal common threshold based on observations from the validation dataset.

Here it is important to note that the $T_S$ constraint in the GEMS model differs from classical temperature-based partitioning methods where the threshold defines precipitation in a binary way as either 100% rainfall or 100% snow. The model simulates snow-precipitation partitioning only until the temperature drops below $T_S$, at which point any precipitation is regarded as 100% snow. For example, when the average temperature (TAVG) is 0°C, using the

assimilated statistical relationships the model will likely simulate some portion of precipitation as snowfall. As illustrated in **Figure 5** at TAVG around of 0°C, the model, on average, simulates around 75% of precipitation as snowfall. Depending on other input variables this ratio varied from approximately 25% to as high as 95%.

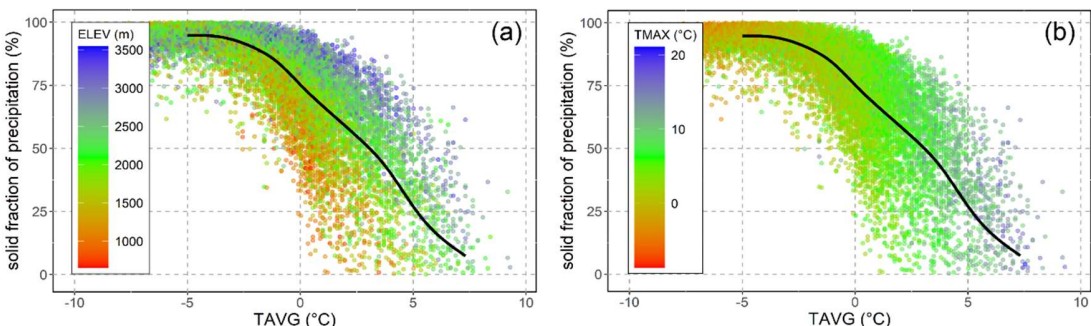


**Figure 5.** The rain-snow transition with respect to average temperature (TAVG) simulated by the pretrained SVR using metadata from the SNOTEL validation data. The two graphs illustrate the same simulations and highlight distributions of elevation (a) and maximum temperature (b). The black line is the median of the resulting solid fraction of precipitation across the simulations.

**4.2  Model evaluation with independent SNOTEL stations**

**Figure 6** presents results of the model evaluation on multi-annual data from the 520 independent SNOTEL stations with histograms and distribution maps of the four selected metrics. The model produces accurate simulations of SWE timeseries in most cases with the median NSE for simulations across all the stations is 0.91, and for 84% of the stations

the model achieved NSE of greater than or equal to 0.8. For 80% of all stations, the maxSWE erorr (maxSWE MAPE) of the simulations is less than 20%, with the median value for all stations being 14%. The median error of the snow meltout date was four days and did not exceed ten days in 74% of instances. We found no spatial associations for NSE values and maxSWE errors, while bias for maxSWE and snow meltout date error tend to be larger in the western part of the study domain (in the vicinity of the Cascade mountains, Oregon state). Here the simulations overestimate

maxSWE and snow meltout date by a larger margin.  Another concentration of overestimation of simulated snow melt-out date occurs in stations located in the Sierra-Nevada mountains. In contrast, the model systematically underestimates maxSWE in some stations in the north-eastern part (Montana and Wyoming), where it consequently simulates earlier snow disappearance.

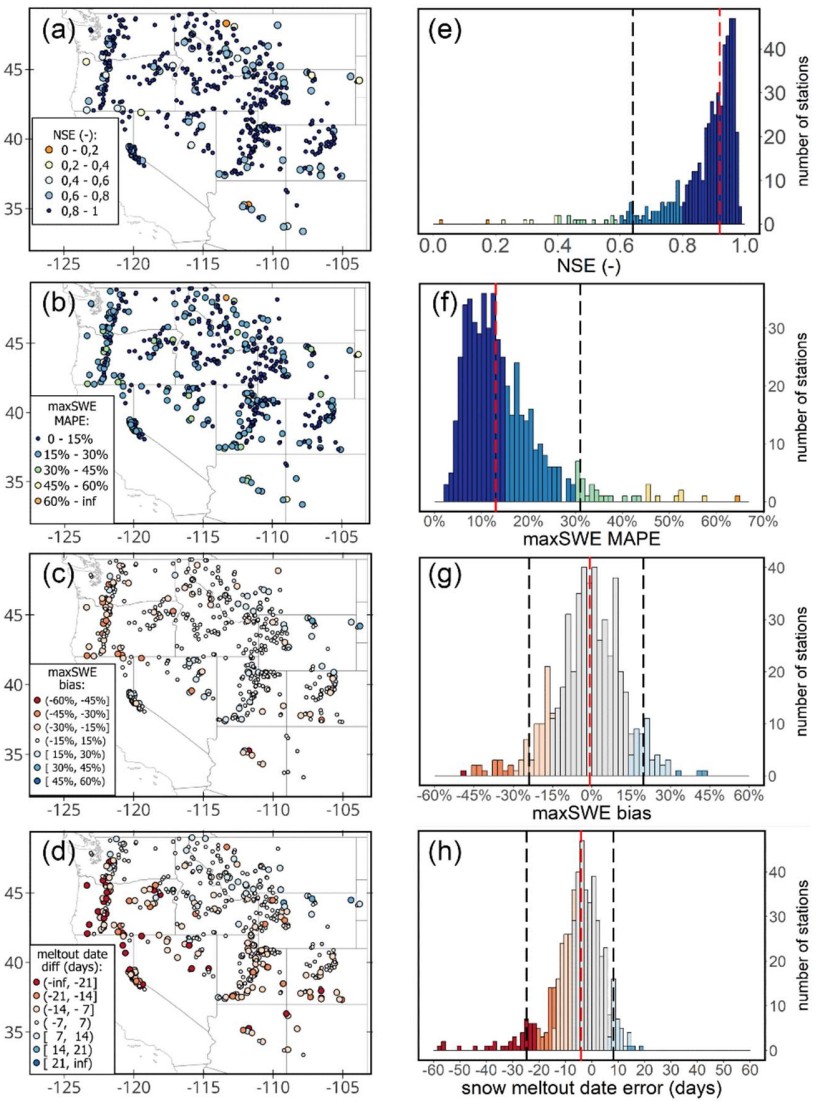

**Figure 6.** GEMS performance metrics for independent SNOTEL stations: spatial distributions of the resultant (a) NSE, (b) maxSWE MAPE, (c) maxSWE bias, (d) snow meltout date error, and corresponding histograms (e-h). Vertical red dashed lines on the histograms denote the median across all stations; the vertical black dashed line correspond to the 5[th] percentile (and 95[th] percentile in case of two-tailed distributions).

In addition to simulations generated with the default $T_S$ value, we also examined the model's accuracy using Ts values calibrated to each SNOTEL station. We calibrated $T_S$ for each of the stations with the objective of maximizing the Nash-Sutcliffe Efficiency of the model`s simulations with respect to observed SWE, and   We bounded the range of calibrated $T_S$ to -5 to +5 °C.  The results illustrated on **Figure 7** show that the station-adjusted modeling incrementally improves all evaluation metrics of the simulations result, though with a lesser impact on mean maxSWE error. Adjusted $T_S$ values tends to be negative across the stations on mountain ranges, particularly across the Cascade and Sierra-Nevada and Rocky Mountains. A cluster of a few stations with positive $T_S$ appear in the northeastern portion of the study region.

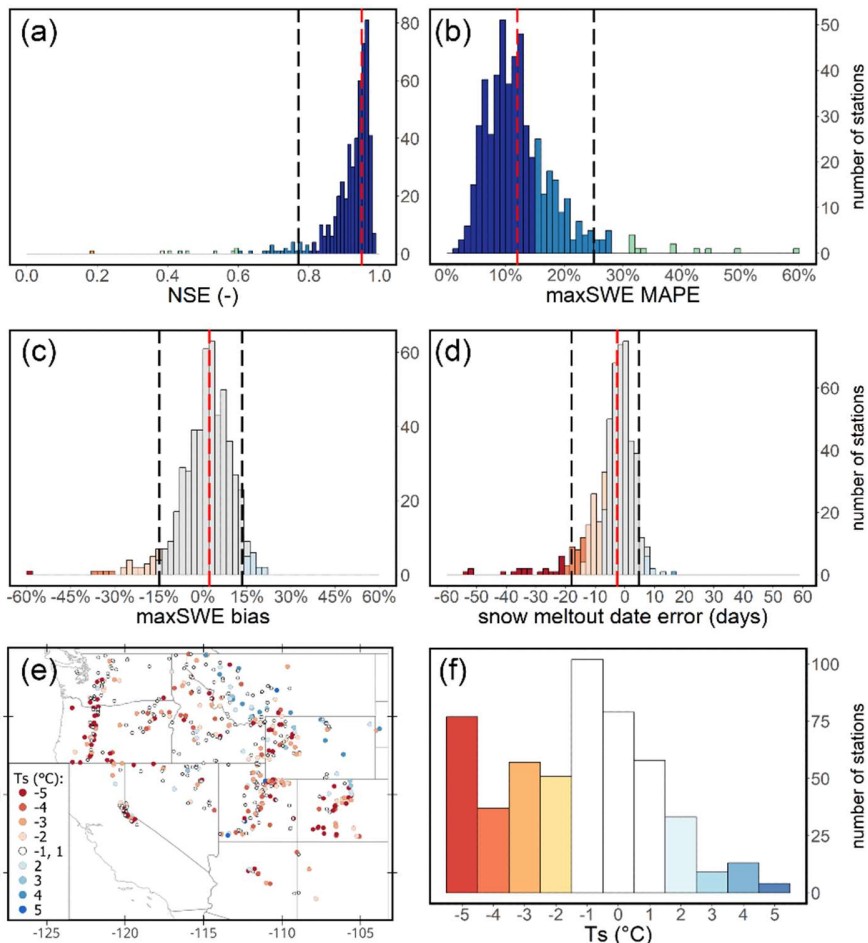

**Figure 7.** GEMS performance metrics for the independent SNOTEL stations with station-adjusted T$_S$ threshold: (a) NSE, (b) maxSWE MAPE, (c) maxSWE bias, (d) snow meltout date error. The bottom map (e) and histogram (f) show density and spatial distribution of the adjusted T$_S$ values.

While the median of the adjusted T$_S$ values for all stations agrees with its default threshold (-1 °C), the density distribution also shows a high frequency of calibrated Ts resulting at the lowest bound of -5 °C (**Figure 7f**). This suggests that, in cases where calibrated Ts values approach the lowest boundary, the model simulations might have been overcalibrated, resulting in error compensation. The overestimation of SWE at these locations can be attributed to several factors that the model does not account for, including effect of dense vegetation, wind induced snow-drift, sublimation, and rain-on-snow events which may be frequent phenomena in the mountain areas (Li et al., 2019; Boniface et al., 2015; Kirchner et al., 2014; Sexstone et al., 2018).

## 4.3 Model evaluation using ESM-SnowMIP reference station data

**Table 4** below presents obtained NSE, maxSWE MAPE and maxSWE bias values of the GEMS simulations using the ESM-SnowMIP reference stations, and the **Figure 8** compares their observed and modelled SWE timeseries. The model performance at ESM-SnowMIP reference sites was robust for the majority of stations with the $T_S$ threshold set to -1 °C by default. In general, simulated SWE was more accurate for the stations located at higher elevations and characterized by higher snow accumulation rates (RME, CDP, WFJ, and SWA), except for SNB, which had the lowest NSE value (0.34) and the highest maxSWE error (17%) among all ESM-SnowMIP stations. The poor performance of the model for the SNB station is attributed to prevalence of wind-induced snow redistribution, which can reportedly reduce peak SWE on the site by up to 40% (Landry et al., 2014). For the same reason, one of the largest SWE errors were recorded for the SNB site by the majority of models that participated in ESM-SnowMIP (Menard et al., 2021).

SWE simulations for SOD and SAP stations have NSE values of around 0.7 and maxSWE MAPE errors of 8% and 18%, respectively. It is important to note that in terms of latitude and thus the range of daylengths, the SOD station is situated much beyond the range of the data utilized to pre-train the GEMS model. In addition, since the Global Continuous Heat-Insolation Load Index (CHILI) does not extend beyond the arctic circle. To estimate it for SOD, we used the nearest known value and assuming flat terrain, but acknowledge that our estimate may have some uncertainty. Regarding the SAP station, GEMS' performance may be affected by the site's anomalous precipitation phase partitioning, in which precipitation reportedly can fall as rain at low temperatures and as snow at temperatures over 5°C (Ménard et al., 2019).

The performance of the model exhibited notable disparities across three forested locations in Canada (OAS, OBS, OJP). In comparison to other sites, the model's performance at these sites was relatively inferior, indicated by NSE values ranging between 0.44 and 0.66 and maxSWE errors spanning from 15% to 30%. This observation suggests a diminished performance of the model in environments characterized by dense canopy interception.

For reference, **Table 4** also provides the NSE of simulations produced by models that participated in ESM-SnowMIP. With the exception of the SNB site, ESM-SnowMIP simulations had lower NSE than those of GEMS simulations. However, a direct comparison between GEMS and ESM-SnowMIP simulations is not possible because evaluation data were not provided to the ESM-SnowMIP participants in advance and rain-snow transitions were prescribed in the driving data (Ménard et al., 2019). ESM-SnowMIP participants thus had no opportunity to enhance model performance by adjusting parameters.

**Table 4.** GEMS performance metrics for the ESM-SnowMIP reference stations.

| Station | **GEMS model** | | | **ESM-SnowMIP models** |
|---|---|---|---|---|
| | NSE | maxSWE MAPE (%) | maxSWE bias (%): | max NSE |
| CDP | 0.84 | 14 | 0 | 0.6 |
| OAS | 0.6 | 15 | -13 | 0.24 |

| | | | | |
|---|---|---|---|---|
| OBS | 0.44 | 29 | -29 | 0.18 |
| OJP | 0.66 | 27 | -30 | 0.41 |
| RME | 0.8 | 13 | -13 | 0.72 |
| SAP | 0.72 | 17 | 3 | 0.47 |
| SNB | 0.34 | 17 | -13 | 0.46 |
| SOD | 0.68 | 8 | 6 | 0.68 |
| SWA | 0.85 | 15 | 14 | 0.6 |
| WFJ | 0.85 | 14 | 4 | 0.64 |

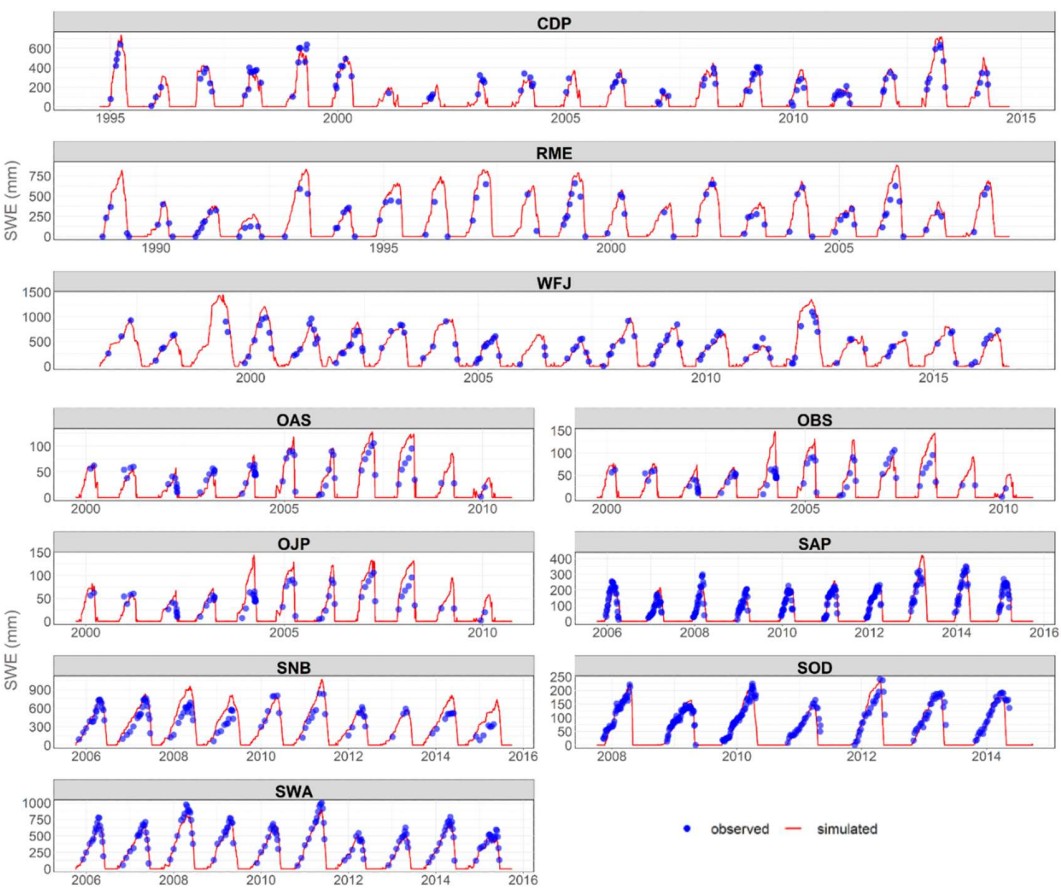

**Figure 8.** Observed and modelled SWE at the ESM-SnowMIP reference stations (with the default $T_s$ threshold)

### 4.4 Model evaluation for large-scale simulations

**Figure 9** compares observed and simulated snow cover area for the selected Western Pamir and Mendoza-Andes regions on daily timestep over two consecutive snow seasons. The primary objective of this analysis was to test and demonstrate the model's transferability to regions with complex terrain and without in-situ SWE data. We assume that if the extent of the simulated SWE aligns well with the remotely sensed snow cover, then the simulated SWE is likely to contain less uncertainty. This assumption is also based on fact that remotely sensed snow cover is increasingly used for parameter calibration or uncertainty reduction in snow modules of hydrological models (e.g. Parajka and Blöschl, 2008; Gyawali and Bárdossy, 2022; Tong et al., 2022; Di Marco et al., 2021).

GEMS accurately reproduced seasonal cycles and interannual variations of snow cover in the Western Pamir and Mendoza-Andes region, which have distinctive seasonal patterns. The simulations capture short-term spikes in the snow cover extent in the middle of the snow seasons over the Pamir. Overall pixel-wise accuracy of snow/no-snow detection for both regions was 92%, while the class-balanced accuracy, which takes into account the balance of class distribution (Branco et al., 2016), was 87% on average.

All validation sites used previously are in the northern hemisphere because we were unable to locate representative station-based snow and climate forcing data for model evaluation in the southern hemisphere. The evaluation of the model in the Mendoza-Andes region implies that the model may have comparable performance for locations in the southern hemisphere.

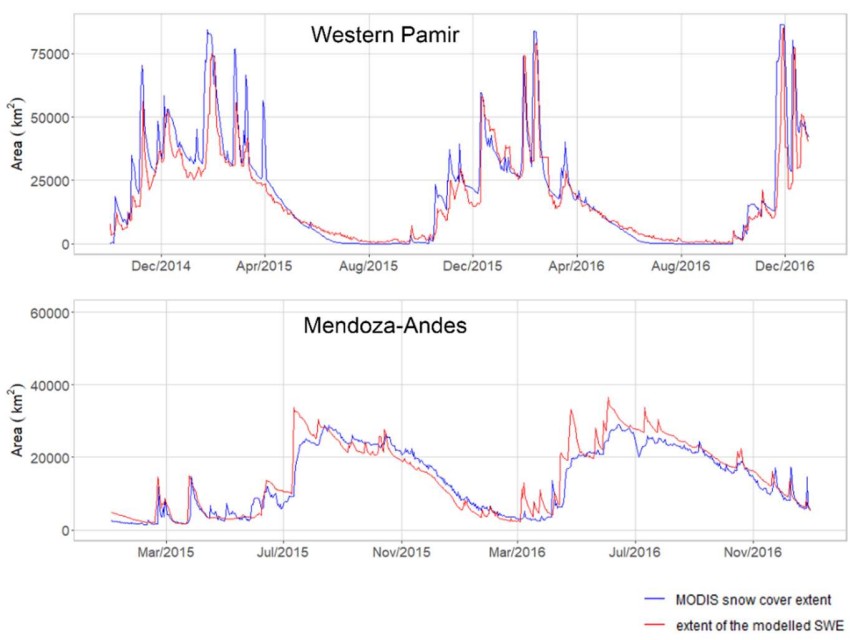

**Figure 9.** Observed and simulated snow cover area for Western Pamir and Mendoza-Andes region

### 4.5 Relative importance of climate and topographic variables

We conducted a permutation-based feature importance analysis to determine how individual input variables affect dSWE simulation. The method randomly shuffles input data and compares the model's baseline performance on the original dataset to performance after permuting a feature's values (Fisher et al., 2018; Greenwell et al., 2018). We applied the permutation-based feature importance analysis on the entire training dataset of the independent SNOTEL

stations as well as its subsamples representing snow accumulation or melt phases. **Figure 10** illustrates the obtained variable importance scores.

The results unequivocally identified precipitation and average temperature followed by daylength as the most significant variables, but they also demonstrate that their importance varies considerably depending on the phase considered. For snow accumulation, precipitation is by far the most obvious and significant variable, followed by a

380 wide margin by maximum temperature. In contrast, the model relies heavily on average temperature and daylength to predict snow melt, followed by precipitation and other remaining variables, again by a wide margin. At first glance, the results suggest that topographic variables are among the least influential, but it should be noted that their significance is assessed in relation to other variables, some of which, such as precipitation and temperature, are more fundamental for accurate snowpack estimation (Günther et al., 2019). Furthermore, climate predictors can be highly

variable, whereas topographic features are constant per each location, which predetermines insufficient variability of these predictors in the dataset and thus contributes to a wider gap of their relative importance in comparison with climate variables.

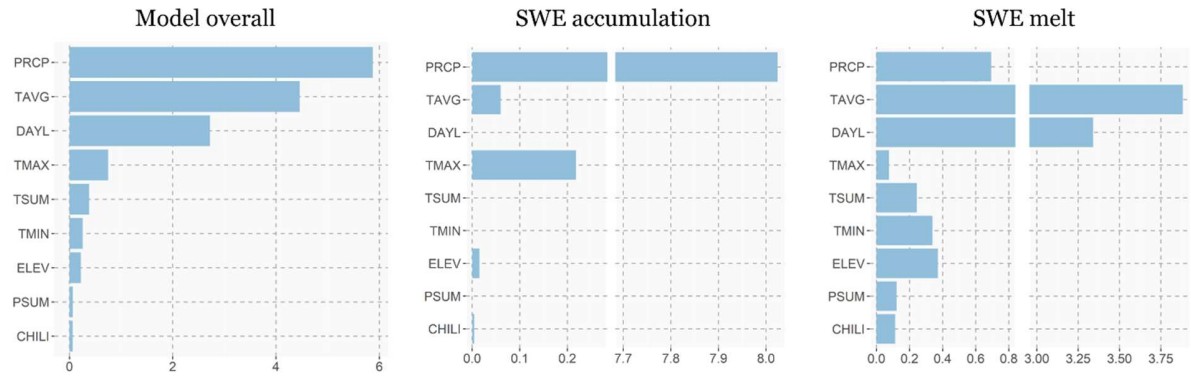

**Figure 10.** Relative importance of model inputs during SWE accumulation and melt phases.

### 4.6 Climatic and topographic attributes of locations where the model exhibited lower accuracy

In order to evaluate model uncertainty, we filtered the SNOTEL stations into two groups based on their NSE values: one group from the 1$^{st}$ quartile, and another group from the 4$^{th}$ quartile of NSE values across all validation SNOTEL stations. We then calculated probability distribution densities for several climatic and topographic characteristics (presented on **Figure 11**) for each group to compare how stations with relatively poorer model performance differ from those with good model performance.

Accordingly, there is a higher likelihood that a station where the GEMS shows relatively inferior performance typically yields lower seasonal snowpack and has higher average seasonal temperatures. In addition, stations with poor model performance tend to have higher diurnal fluctuations during the snow season. We have detected minor differences between two groups of stations in terms of average elevation or distribution of heat-insolation indices. Despite not using station latitude in the model as a direct input (it is required to estimate daylength for the location), the comparison suggests that there was a slightly higher proportion of poorly performing stations at lower latitudes.

These distinguishing characteristics of poorly performing stations in some instances are not mutually exclusive. For example locations with higher seasonal temperatures usually tend to have lower seasonal SWE peaks under identical conditions. Similarly, lower latitudes in the western US have generally greater diurnal air temperature variations. We hypothesize that the performance of the model under such climatic conditions could be enhanced by incorporating more respective observations into the training dataset, which apparently included fewer SNOTEL stations from the southern part of the training domain (see **Figure 2c**).

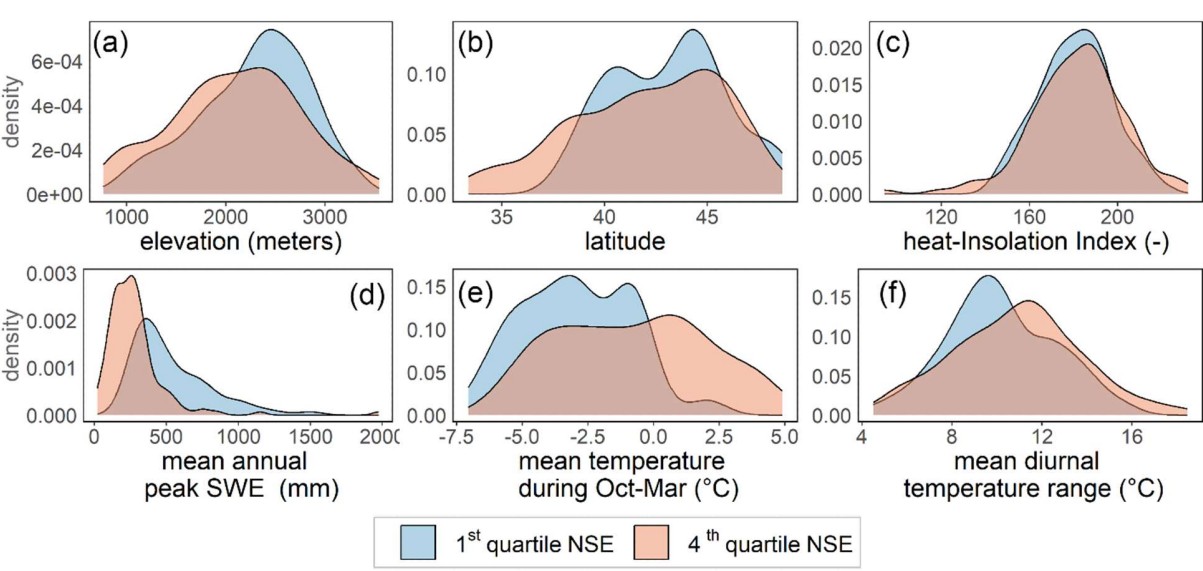

**Figure 11.** Probability density distributions of topographic and climatic characteristics of the SNOTEL stations where the model shows higher and lower performance in terms of NSE.

## 4.7 Performance of GEMS model under different input requirements

To evaluate how the model performance depends on a number and type of input data (see **Table 1**), we compared simulations of the three versions of GEMS using the SNOTEL validation dataset. Overall, the incorporation of diurnal temperature range and heat-insolation index enhances simulation accuracy as measured by a smaller interquartile range of NSE and maxSWE error (**Figure 12**). Compared to utilizing only the maximum and minimum temperatures, the heat-insolation index is a predictor that appears to modestly improve model accuracy. This improvement is evident, as compared to GEMS-6P, GEMS-5P exhibits somewhat better performance across the four metrics used.

Besides, GEMS-7P and GEMS-5P have a tighter range between the minimum and maximum NSE and maxSWE error. However, there is no discernible difference in the snow meltout date and maximum SWE bias across the three model versions. Although GEMS-4P has a slightly lower NSE and maxSWE error accuracy, its overall performance is still robust and it has the benefit of requiring less inputs (only precipitation, average temperature, elevation, and latitude).

Running any of the three versions of the model on a desktop computer using single CPU core (Intel i7) took less than 6 seconds for 20-year long Weissfluhjoch station data, which approximates to 0.3 seconds per site-year. An ongoing experiment (not shown here) suggests that the computation time can be reduced by about 30% through improved sampling of the training data used to develop the model, a modification that will be implemented in the updated version of GEMS.

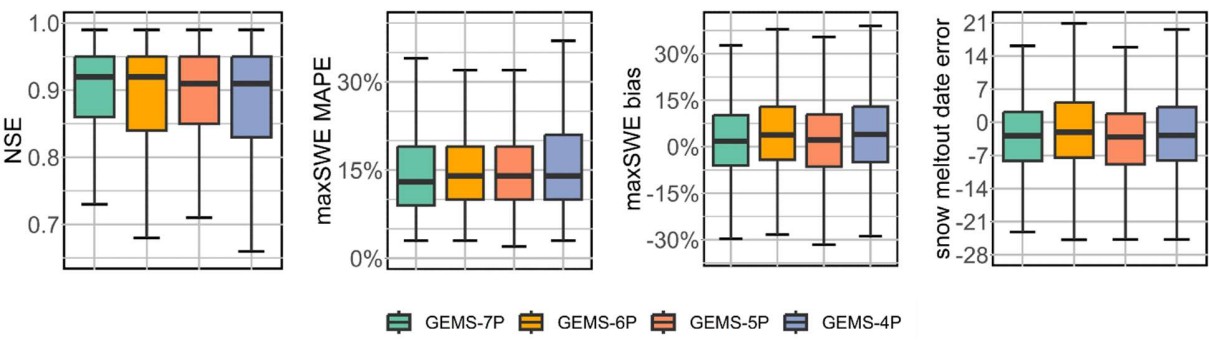

**Figure 12.** Comparison of performance across three GEMS models. The boxplots` minimum and maximum limits correspond to $1^{st}Q-1.5*IQR$ and $3^{rd}Q+1.5*IQR$ respectively.

## 5. Model limitations

Instances of less consistent simulations generated by the model can arise from various sources of uncertainty, including internal uncertainty within the model, as well as uncertainty in input data and unaccounted external factors.

One of the major limitations of the model is that it does not account for vegetation, which is known to have a complex and divergent effect on snow accumulation and melt under different climate conditions (Dickerson-Lange et al., 2021;

Sun et al., 2022). Because most SNOTEL stations are situated in forest clearings or open bushes, we initially assumed the training sample locations to be free of canopy obstruction. A visual check using Google Earth of the stations used in both the training and validation samples reveals, however, that some sites can be intercepted by the tree canopies in their surroundings. In addition, we have detected the shadowing of some snow pillows by the dense forests that surround them. Both phenomena are possible sources of model uncertainty, also evidenced from relatively lower performance of the model for three forested ESM-SnowMIP sites in Canada (OAS, OBS, OJP), and future model development should try to incorporate vegetation effects.

Based on a comparison of high performing and low performing site simulations, the model may be less accurate at simulating shallow snow in warm sites at lower elevations. When these factors combine with large diurnal temperature fluctuations, model simulations may even become more distorted. These issues could be resolved with a more sophisticated sampling strategy and by incorporating additional observations into the training of the model. It remains unclear however, whether an improved sampling strategy could also better approximate rain-on-snow effects, as these are driven by dynamic processes of energy exchanges across snow layers that the model does not capture for.

The model's parsimonious design, which relies only on precipitation and temperature variables as climate data inputs, also precludes the incorporation of wind- or gravity-induced snow redistribution, which may compromise the accuracy of single-point simulations for wind-exposed sites.

When temperature is below the -1°C Ts threshold and precipitation is zero, GEMS will automatically estimate daily change in SWE as 0 mm. The model thus fails to account for snow sublimation, which can occur even when temperatures are below freezing. This differs from snow models based on energy balance, which can estimate snow sublimation. Furthermore, the evaluation on the SNOTEL dataset suggests that significant adjustments of the Ts threshold imposes a risk of error compensation due to over-calibration. Therefore, we recommend adhering to the default value of Ts (-1°C), unless local precipitation-snow partitioning patterns are well understood.

As discussed in the Section **4.6** and also evidenced from the evaluation on ESM-SnowMIP sites, the model demonstrates relatively better performance in mountainous areas compared to lower elevations. However, the training dataset used to elaborate the model may be less representative of locations with very low CHILI indices (**Figure 2d**). Low CHILI indices often correspond to sites significantly shadowed by terrain or situated at higher latitudes or both. This discrepancy may be an additional source of model uncertainty.

## 6.  Summary and conclusions

We present a computationally efficient model that emulates snow mass accumulation and melting using only a few climate and topographic inputs. The absence of the explicit need for calibration is the most distinctive aspect of the GEMS model, with 100% rain-snow transition temperature threshold ($T_S$) being the only parameter that can be modified, though in most validation cases, robust simulations were obtained just using the default $T_S$ value. Despite

its parsimony and no extensive calibration options, the model achieves robust transferability across a variety of climate and geographic conditions.

The main motivation behind the development of GEMS was to balance the trade-off between complexity, data requirement, and transferability, which can be helpful for operational monitoring and hydrological modelling in data scarce domains. The emulator was developed by training a machine learning model on daily changes in snow water

equivalent as a response to daily climate inputs and diverse topographic features. Despite the dynamic nature of snow processes, our simplified "static" approach effectively captured the impact of precipitation, temperature, and topography on snow melt, as indicated by the validation results. This corroborates the conclusion of several intercomparison studies that model complexity is not necessarily a predeterminant of its performance (Essery et al., 2013; Magnusson et al., 2015; Menard et al., 2021).

The model evaluation suggests that GEMS achieves comparable performance to physical snow models, as evidenced by comparing with simulations from ESM-SnowMIP. A more appropriate comparison might necessitate adjustment of physical model parameters, which was not investigated in ESM-SnowMIP. Nevertheless, the evaluation outcomes allow us to conclude that, at the very least, GEMS with its default $T_S$ parameter exhibits superior spatial transferability compared to physical models with unadjusted parameters.

In addition to avoiding computationally demanding calibration, GEMS may also help to address the equifinality of model parameters that is pertinent to hydrological and snow modelling. The challenge of equifinality is particularly pronounced in hydrological modeling, where even relatively simple snow models require calibration of at least two parameters: the precipitation-snow threshold and the degree-day melt factor. Considering that there are many other parameters for the remaining components of a hydrological model, it is easy to end up with multiple combinations of

optimal parameters. In contrast, GEMS shows generally plausible performance in diverse climatic and topographic conditions using the default value of $T_S$.

One difference between GEMS and physics-based models lies in the number of outputs they generate. While GEMS is specifically designed for simulating only SWE, comprehensive physics-based snow models produce a broader spectrum of outputs that provide valuable insights into other snow properties. We assume that machine learning could

become helpful in modelling some of these snow properties. For example, previous studies have shown how simple empirical models can effectively derive snow depth from SWE measurements and vice versa (Aschauer et al., 2023; Hill et al., 2019). We assume that a similar approach to GEMS could be scalable for estimating snow depth by incorporating additional variables, such as snow age.

Machine learning is gaining importance in snow modelling, with existing applications predominantly focusing on

snowpack interpolation or the detection of its instantaneous state through the assimilation of ground-truth and active satellite radar data. GEMS provides a modelling framework similar to traditional snow modelling approaches, by simulating snowpack in a temporally progressive manner and leveraging climate and topographic inputs commonly used in snow models. Moreover, the revealed variable importance aligns with the general physics governing how climate variables affect snowpack accumulation and melt. Some recent studies employing machine learning methods

(Vafakhah et al., 2022; Duan et al., 2023) also simulate snowpack in a temporal manner and demonstrate robust performance, though spatial extrapolation limits of those algorithms remain unclear. Another recent study (Wang et al., 2022) presents promising results for a deep learning-based approach, showcasing its superior spatial transferability compared to enhanced temperature index model across the United States. Nevertheless, the applicability of these models beyond their targeted regions may be questionable due to dependance on climate inputs or locally-specific data that may not be available elsewhere. From these perspectives, GEMS offers a higher degree of parsimony in terms of required input variables and, more importantly, a proven ability to generalize outside of the training domain.

We have tested several other data-driven techniques for the model development, including multivariate linear regression, Gaussian process, Random Forests, and Gradient Boosting Machines (not shown here). When evaluating on the training dataset, the performance of most models was either lower or equivalent to SVR; however, even in the latter case their accuracy on the evaluation dataset was worse. Experiments in other fields indicate that SVR has relatively better extrapolation potential on unseen data (Horn and Schulz, 2011; Kim and Kim, 2019), which may explain why it outperformed other algorithms. We have not examined neural network algorithms since they take more computer resources during training, and evidence suggests that they tend to underperform relative to other machine learning ML techniques when applied to tabular data (Borisov et al., 2022; Shwartz-Ziv and Armon, 2022). To make definitive judgments with regard to performances of different machine learning algorithms, however, would require a more extensive intercomparison experiment which is outside the scope of this paper.

Future development of GEMS may aim at addressing vegetation effects and improving model performance for shallow snowpack in warm sites. Including sublimation and rain-on-snow effects may be possible but though inevitably lead to increased complexity of the model. Another promising aspect of model improvement involves further reduction of computational costs. At least to some extent, these improvements may be achieved through a more careful selection and sampling of the training dataset used to develop the model. In addition to these imperatives, further work may also concentrate on extending the similar framework for modelling other snow properties, such as snow depth and albedo.

**Code and data availability**

The exact version of the GEMS model used to produce the results used in this paper is available on https://zenodo.org/doi/10.5281/zenodo.7929178 under the Creative Commons Attribution 4.0 International license. The model repository also contains files with a validation set of temperature-bias-corrected SNOTEL data, as well as ESM-SnowMIP stations data aggregated to daily time scales. The original SNOTEL data is accessible via https://wcc.sc.egov.usda.gov/reportGenerator (USDA, 2022). The original ESM-SnowMIP reference station data is accessible at https://doi.pangaea.de/10.1594/PANGAEA.897575 (Menard and Essery, 2019). CHELSA-W5E5 v1.0 data is accessible at https://data.isimip.org/10.48364/ISIMIP.836809.3 (Karger et al., 2022b).

**Author contributions:** AU and DM designed the study, AU performed computations, RE provided feedback on model evaluation. All authors contributed to writing and review of the manuscript. DM supervised the project.

**Competing interests:** The authors declare that they have no conflict of interest.

**Acknowledgments:** This research has been supported by the Volkswagen Foundation within the 'Structured doctoral programme on Sustainable Agricultural Development in Central Asia' (SUSADICA) project, Grant Number 96 264.
Publication of the manuscript was supported by the GEO Mountains Initiative within the project "High-resolution daily snow reanalysis dataset for Tian-Shan and Pamir mountains (1980–2016)". We would like to thank Dr. Matthieu Lafaysse and an anonymous reviewer for their valuable comments and suggestions. We would like to acknowledge the use of IAMO`s computing facilities in this research. We extend our sincere appreciation to the open source developer community, and individuals behind numerous R packages, including but not limited to "e1071" (Meyer et
al., 2023), "terra" (Hijmans, 2023), "hydroGOF" (Mauricio Zambrano-Bigiarini, 2020), "vip" (Greenwell and Boehmke, 2020), and "dplyr" (Wickham et al., 2023).

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
