# Peer review of "GEMS v1.0: Generalizable empirical model of snow accumulation and melt based on daily snow mass changes in response to climate and topographic drivers"

_Geoscientific Model Development, 2023_

## Author Comment (AC1)

Dear Dr. Matthieu Lafaysse,

Thank you much for your review and valuable comments. Below you will find your referee comments (in black) and our responses (in blue).

With regards,

Atabek Umirbekov, on behalf of all authors

**General comments**

Umirbekov et al. present a new machine learning approach to simulate snow mass with parcimonious data input and an extremely low numerical cost. The evaluation framework is really interesting as it includes independent data removed from the calibration dataset, but also the state-of-the-art ESM-SnowMIP dataset including challenging climate and environment conditions beyond those of the calibration dataset, and finally a spatialized application with more uncertain forcing data and evaluation data derived from remote sensing. Of course, the potential of machine learning has to be considered in snow modelling and I think this paper can be a significant contribution on that topic. The results clearly challenge physical models, even if obviously the output variables are not sufficient for all applications.

Nevertheless, I think the description of methods and results is sometimes a bit too fast in the current version of the manuscript and that some details are missing for an accurate understanding and interpretation of results. In general, figures are not really introduced in the main text. I would also have expected more in-depth discussion of the advantages and disadvantages of this approach compared to physical approaches and other machine learning approaches in the light of presented results and previous literature, and also discussions about the possibility to disentangle errors due to the forcing and to the algorithm itself. Maybe, the chosen structure of the paper that mixes results description and results discussion is partly responsible for this sometimes incomplete discussion. Finally the choice to try to recalibrate the Ts parameter is sometimes confusing especially when it's done on evalution datasets, as it leads to unrealistic values and overcalibration.

I also have some specific comments or questions below that can probably be addressed rather easily by the authors during the revision process.

We appreciate your overall feedback on the manuscript and are grateful for your valuable comments. In response to your suggestions, we will incorporate a comparison with physics-based snow models using proceedings from the SnowMIP2 study (Krinner et al., 2018), and include a discussion of the advantages and disadvantages of the model in comparison with other physical snow models. This discussion will be also complemented with comparisons to other machine learning applications in snow modeling. We will introduce all figures in the main text for better context. Furthermore, we will restructure the manuscript to ensure that results and their corresponding discussions are appropriately organized in their respective sections. Finally, we will provide a more detailed description in the main text regarding the precipitation-snow partitioning and the 'Ts' parameter to prevent any confusion with traditional temperature-based partitioning methods.

**Detailed comments**

Section 2.1 The choice of SVR relatively to other machine learning algorithms is not discussed. I would suggest to add a quick summary of advantages and disadvantages compared to the most classical algorithms available in literature (random forests, convolutional neural network, simpler regressions, etc.)

Thank you for this suggestion. We will enhance the introduction by incorporating a brief summary of machine learning application to snowpack modeling. In addition, we suggest to include a brief paragraph into the discussion section summarizing some our experiments and hypotheses. This passage would include (but might be not limited to) the followings:

 *"We have evaluated several other data-driven techniques for the model development, including linear regression, Random Forests, Gradient Boosting Machines, Gaussian process (not shown here). When compared to the training dataset, the performance of most models was equivalent; however, their accuracy on the evaluation dataset was worse. Experiments in other fields indicate that SVR has relatively better extrapolation potential on unseen data (Horn and Schulz, 2011; Kim and Kim, 2019), which may explain why it outperformed other algorithms. We haven't examined neural network algorithms since they take more computer resources during training, and evidence suggests that they tend to underperform relative to other machine learning ML techniques when applied to tabular data (Borisov et al., 2022). To make definitive judgments in this regard, however, would require a more extensive intercomparison experiment that is outside the scope of this paper."*

Can you define more explicitly i, j, N, xi, xj, X ?

We will incorporate more explicit notations for the variables and parameters denoted in the main formula (1).

I understand from Fig.1 and Eq. 2 that when temperature is below the -1°C threshold and precipitation is zero, then dSWE is always equal to 0. Is that correct ? How often does this assumption fail in the training or evaluation dataset? Does this imply an intrinsic limitation of GEMS for transferability on steep slopes where the surface energy balance can be positive even at negative temperatures ? (I think it does.)

Thank you for these guiding questions. Yes, when temperature is below the -1°C threshold and precipitation is zero, dSWE will be automatically set by the model code to 0 mm. Due to the temperature threshold and not using/estimating solar balance, the model does not capture snow melt and sublimation which. We recognize this as one of the model limitations (particularly compared with most energy-balance snow models). We will note this limitation in the respective part of the manuscript accordingly.

Section 2.2

The authors say they « fine-tuned the hyperparameters so that the model produces similar levels of accuracy when applied to observations from the same stations for 2019 and 2020. » I understand the general idea but the detailed procedure is not accurately described. Can you describe the detailed protocol for this « fine-tuning » ?

We will complement the respective lines of the manuscript with the following details:

 *"The hyperparameter calibration process involved an exhaustive 'grid-search' technique, which systematically explored all possible combinations within predefined parameter ranges. Ultimately, we selected the hyperparameter configurations that resulted in the lowest root mean squared error between simulated and observed dSWE during both model training on observations from 2017-2018 and its testing on observations from 2019 and 2020."*

As solid precipitation measurements are prone to large measurement errors and is one of the main predictor of the model, I would have expected more details about precipitation gauges used in the SNOTEL network, procedures applied to account for undercatch, and if possible estimated uncertainties.

We suggest to include the following passage into the Data section:

*"SNOTEL stations primarily utilize tipping bucket-type precipitation gauges, which are reported to have an accuracy range between 2% and 5%, depending on the type (USDA, 2010). It is also recognized that precipitation gauges are susceptible to solid precipitation undercatch, especially when snowfall occurs in windy conditions (USDA, 2014). Scalzitti et al., 2016 provide a comprehensive review of the issues associated with precipitation undercatch, highlighting reported undercacth ranging from 11% for snowfall under 2m/sec wind speed to more than 30% during intense snowstorm events."*

Section 3 I think « Model evaluation » would be a more appropriate title than « model validation » as a model can never be considered as fully validated.

Thank you for this suggestion. We agree and will change the section to "Model evaluation".

The authors say « we excluded stations that exhibit precipitation undercatch, which we formulate as when SWE accumulated by March is greater than the accumulated precipitation during October to March. ». I would expect all stations to be affected by precipiton undercatch and total SWE to be always higher than raw precipitation measurements. Do you apply a specific threshold to only eliminate major undercatch ? Or do you use precipitation timeseries that are already corrected for precipitation undercatch following WMO recommendations ? My misunderstanding is probably linked to the lack of details in Section 2.2 as previously mentioned.

Then, was this selection procedure also apply to the training dataset ? If not, why ?

We appreciate these comments and questions. We agree that this part needs more clarifications, and suggest to include some additional explanatory exerts. These may include (but are not limited to) the following:

*"While SNOTEL stations may be susceptible to precipitation undercatch, especially during intense snowfall events and high winds (Scalzitti et al., 2016), it is essential for machine learning to have accurate training data. To ensure data accuracy, we cleaned the training dataset by removing observations with inconsistencies between daily precipitation and snow mass accumulation. These inconsistencies refer to cases when the daily increase in SWE exceeded the reported daily precipitation.*

*The selection approach differed for the evaluation dataset because we aimed to retain as many stations as possible for evaluation and besides that the model requires complete daily time series without missing observations. We therefore used aggregated*

*precipitation sums from October to March to compare with accumulated SWE by March. This approach enabled us to include more stations in the evaluation dataset while excluding only those hydrological years that exhibited inconsistencies between these variables. When using this filter, we did not set any specific threshold for the magnitude of inconsistencies, nor did we make corrections to the precipitation time series."*

It should be also noted that we included a criterion that required at least five hydrological years of observations for a station to be part of the evaluation dataset (line 150). Consequently, some SNOTEL stations were excluded based on this specific requirement.

Section 3.1

L193 I would suggest to start by a sentence presenting the Figure before providing its interpretation.

Yes, we will introduce the Figure in the text before its interpretation.

In Figure 4 « actual » should be replaced by « observed ». Is there a reason to present the simulations in the X axis and not in the Y axis (that would be more common for a scatter plot) ?

Thank you for pointing at this. We will replace 'actual" with "observed". We will also swap current X and Y axis accordingly.

In Figure 5, it is not immediate to understand what is represented because the caption is not self-sufficient and the description in the text is also too vague. The definition of TAVG should be remind in the caption. Then what does represent a single point ? A station and a date ? Then, this solid fraction of precipitation does not really appear in model description, neither in Figure 1 neither in the Equations, so it is difficult to understand how this diagnostic is obtained from the provided model description. The reason for providing this Figure is also unclear as finally these outputs are not really used as a fixed temperature threshold finally replaces the values obtained by the algorithm. This needs to be clarified.

We appreciate your suggestion to incorporate additional clarifications into this section. We will add more details under the Figure caption. Furthermore, we will add more clarifications into the text, such as the following:

*"The Ts constraint differs from classical temperature-based partitioning methods where the threshold define precipitation in a binary way as either 100% rainfall or 100% snow.*

*The model simulates snow-precipitation partitioning using its inherent learned algorithm, but only until the temperature drops below Ts. At that point, any precipitation is regarded as 100% snow. For example, when the average temperature (TAVG) is 0°C, the model will likely simulate a portion of precipitation as snow. As an example, as illustrated in Figure 5 at TAVG at around of 0°C, the model is likely to simulate a significant portion of precipitation as snow (around 75% of precipitation) even if Ts is -1°C."*

As for the other Figures, introducing quickly Figure 6 would be helpful before providing the results analysis. In the description of the results of Figure 6, detailed references to the subplots would help to follow results description.

We will introduce the Figure 6 in the text.

Isn't the maxSWE score more representative of the quality of input precipitation than of the skill of the SVR model ?

Yes, given the temperature threshold, we assume that maxSWE might be more representative of precipitation input accuracy. However, since a portion of the simulated maxSWE is influenced by the model's simulation of dSWE (at temperatures above the Ts threshold), we think it is reasonable to keep maxSWE as one of the metrics.

L252-254 If removing stations with incorrect measurements is understandable, removing stations with snow drift should be avoided as snow drift is not a measurement error, it's a natural process challenging to reproduce with physical models and also maybe with machine learning models, but the general ability or unability of any model to reproduce snow conditions should account for places where snow drift happen.

Thank you for raising this concern. Instances where recorded maxSWE exceeds accumulated precipitation may be due to snow-drift, precipitation undercatch, or a combination of both factors. Unfortunately, attributing these inconsistencies to individual factors may require a separate research effort. We therefore had to exclude those stations from the evaluation, but we note the inability of the model to capture snow-drift in line 263-265 and explicitly state this as a model`s limitation in line 407-408.

L255-256 You mean that an overcalibration is obtained due to error compensation between snow drift and rain-snow transition ? Could the sentence be more clear ?

Thank you. Yes indeed, we meant that overcalibration may lead to error compensation. We will make it more explicit in a new version of the manuscript.

Section 3.3

Again, an introduction of Figure 8 in the text would be useful.

We will introduce the Figure 8 in the text.

L266-267 It is not obvious which value of NSE should be considered as « acceptable ». Indeed, NSE is easily high when dealing with variables with a high seasonal cycle. What would be the NSE value of the daily interannual mean of observed SWE ? Is the 0.7 value at Sapporo better than such a reference score ?

We intended to refer to some categorizations of NSE across multiple studies (e.g. N. Moriasi et al., 2007). However, we recognize that these classifications, designed for hydrological models, might not be directly applicable for classifying snow model outputs. Therefore, we will revise sentences with qualitative classifications like this one.

L269-270 This could be moved to the Method section

We agree with and will implement this suggestion.

L274-280 As it was already noticed with the SNOTEL dataset that local calibration of the Ts threshold leads to severe error compensations, and as the purpose of the application of the GEMS system on the ESM-SnowMIP dataset is to assess its spatial transferability beyond its training dataset, I am not really convinced of the interest to test again to recalibrate locally this threshold on each ESM-SnowMIP site. The conclusions that again this leads to overcalibration and errors compensations were rather expected, so I would suggest to remove thisanalysis.

Thank you for these insights and the suggestion. We acknowledge that calibrating the Ts threshold may result in error compensations. However, the results do not provide insights into the extent of these compensations. As previously described, the Ts threshold in the model differs from the classical temperature-based threshold method. For instance, when Ts is set at -3°C and temperature (TAVG) is 0°C, the model will likely classify a larger portion of precipitation as snow (Figure 5). Nevertheless, we recognize that calibration in general might be inappropriate when assessing the spatial transferability of the model. Hence, we ´will remove the calibration analysis for Snow-MIP stations from the manuscript.

Apart from model evaluation, calibration could still be useful during model application, particularly when local precipitation-snow partitioning patterns are known. In light of this, we suggest the following: 1) more explicitly acknowledge the risk for error compensation due to calibration in the model limitations section, and 2) recommend adhering to the default value of -1°C unless the local precipitation-snow partitioning patterns are known."

Section 3.4

Again an introduction of Figure 9 is missing.

We will introduce Figure 9 in the text.

My feeling is that the level of discussion in this section is not as advanced as for the evaluation on ESM-SnowMIP sites. How does this skill in terms of snow cover extent compare with physical models ?

Thank you for this suggestion. We acknowledge that this section's content is not as comprehensive as other sections, particularly in terms of comparison with the performance of physical models. However, the extensive computational burden for such a comparison present a significant challenge to us.

The primary objective of this section is to test and demonstrate the model's transferability to regions with complex terrain and lacking in-situ SWE data. We assume that if the extent of the simulated SWE aligns well with the remote-sensed snow cover, then the simulated SWE is likely to contain less uncertainty. This assumption is also based on fact that *remote sensed snow cover is increasingly used for parameter calibration or uncertainty reduction in snow modules of hydrological models (e.g. Parajka and Blöschl, 2008; Gyawali and Bárdossy, 2022; Tong et al., 2022; Di Marco et al., 2021)."*

We would like to suggest to elaborate on these points in the new version of the manuscript.

Section 4.1

L312 Reference error.

This was intended as a reference to Figure 10. We will correct this in a new version of the manuscript and appropriately introduce Figure 10 in the text.

L315 Could the relatively low contribution of the heat-insolation index be possibly explained by an unsufficient variability of this predictor in the training dataset ?

Yes, this is what we intended to state. We will revise the sentence making this clearer.

In mountainous areas, shadows and slope inclinations are a major factor to explain melting. But I assume that all observations correspond to flat areas, and maybe the variability of shadows in the SNOTEL network is neither representative of the variability of topographic conditions in mountains. This is important to discuss as it could limit the possibility to apply this algorithm on areas with complex topography.

We appreciate these comments and suggestions. Indeed, the SNOTEL stations utilize flatbed pillows, but are primarily situated in mountainous regions. As discussed in Section 4.2 and illustrated in Figure 11, the model demonstrates relatively better performance in mountainous areas compared to lower elevations. However, when examining the histogram of CHILI values in Figure 2, it becomes apparent that the training dataset may be less representative of locations with lower CHILI indices. These lower indices often correspond to sites significantly shadowed by terrain or situated at higher latitudes or both. This discrepancy poses an additional potential source of model uncertainty, a point we will further discuss in the section on model limitations.

Section 4.2

I am wondering how much this conclusion is affected by the choice of NSE to quantify errors. Indeed, as this score is highly influenced by the existence of a seasonal cycle, it is rather normal to get better scores with deeper snowpacks that exhibit a very strong seasonality than on sites with more intermittent snow cover. Considering other scores (for instance a Root Mean Square Error), I would not be surprised that stations with the poorest performance would be reversed. Can you comment on that topic ?

Thank you for your guiding questions. We agree that NSE alone may not adequately distinguish between cases of 'good' and 'poor' model performance, and use of different metrics would likely result in varying compositions of these two performance groups. We assume that Root Mean Square Error may serve as a more suitable alternative for comparing the model performance across the stations. Consequently, we suggest to revisit the analysis in this section using RMSE (probably comparing lowest and highest quartiles) and incorporate it as an additional metric in the evaluation section.

L375 The authors say that « GEMS also addresses the equifinality issue that is pertinent to hydrological and snow modelling. » but the only parameter they have introduced (Ts threshold) clearly raises a very strong equifinality resulting in possible overcalibration to compensate various possible errors including snow drift, precipitation undercatch, etc.

We assume that this sentence is now justified, considering the preceding explanation of how Ts works in the model, how it differs from temperature-based partitioning methods, as well as our intention to stick to the default Ts in our recommendations.

In this sentence we refered to the challenge of calibrating multiple parameters in hydrological and snow modelling. This challenge is particularly prominent in hydrological modeling, where even relatively simple snow modules require calibration of at least two parameters: the precipitation-snow threshold and the degree factor. Considering that there are many other parameters for different components of a hydrological model, it would be easy to end up with multiple combinations of optimal model parameters. We hypothesize that replacing the snow module with a model that is based on generalizable empirical relationships may help to reduce the equifinality issue, especially when employing conceptual hydrological modeling.

L388 « GEMS can, for instance, provide information for the parameterization of physics-based models, e.g. precipitation phase partitioning and its elevational dependence ». I don't see how the results presented here suggest this conclusion and considering the strong risk of overcalibration of this Ts value (leading to clearly unrealistic values below -5°C), I am not convinced at this point that GEMS could help me to discriminate between snow and rain.

As mentioned earlier, we acknowledge that calibrating Ts poses a risk of error compensation, though considering how Ts operates in the model, the extent of overcalibration maybe not as pronounced as it would be with traditional temperature-based thresholds. Despite this, we recognize that the statement in this sentence may have been too assertive and requires further verification. We will remove this sentence from the manuscript.

There is a section 5.1 but not any section 5.2. Maybe a subtitle for the first part of Section 5 is missing.

As it was also recommended by Reviewer 2, we would like to separate section 5 into two separate sections in a new version of the manuscript: section 5 'Model Limitations' and section 6 'Discussion'.

L393-400 The authors discuss the limitations of their approach relatively to forest areas but they seem to have intentionnally remove the 3 forest sites of the ESM-SnowMIP dataset from their evaluations. This should at least be discussed if there is a valid reason for that. But even if the model skill is lower on the 3 Canadian forest sites, I would have included these sites in the evaluations to provide concrete results to support this discussion.

Indeed, we haven't evaluated the model on the three Canadian sites but because at that time we couldn't precisely locate the sites to determine CHILI parameters. We will include these sites for the model evaluation in the next version of the manuscript.

L408-410 Unfortunately, blowing snow can be an important process even at large scale especially in polar regions. So large scale applications of the system may still be affected by this limitation.

We will remove that part of the sentence.

The discussion do not compare the skill of this approach with the skill of physical models while similar metrics are provided at the same sites in Ménard et al., 2021, and other evaluations are also available in the literature for snow cover extent. I think this would be important to consider as well.

We appreciate this suggestion. We will compare the skill of the model with that of physical models that participated in ESM-SnowMIP, using model simulations presented in Krinner et al., 2018.

The discussion or final summary also lack comments about the strengths and weaknesses of their results compared to the literature cited in the introduction applying machine learning to predict snow mass.

We will present our perspective on the strengths and weaknesses of our model approach compared to other cases of snow models utilizing machine learning.

Furthermore, the outputs of the model are currently limited to SWE while several snow-sensitive applications require more variables (e.g. surface temperature for NWP and climate modelling, snow internal properties for remote-sensing retrieval

algorithms or avalanche forecasting). This limitation should also be mentioned with possibly discussions about the feasability to extend this approch to more variables.

Thank you for this suggestion. We will include this as a limitation of our model and complement it by presenting our perspective on the snow processes to which our approach may be applicable.

**References:**

Borisov, V., Leemann, T., Seßler, K., Haug, J., Pawelczyk, M., and Kasneci, G.: Deep Neural Networks and Tabular Data: A Survey, IEEE Trans. Neural Networks Learn. Syst., 1–21, https://doi.org/10.1109/TNNLS.2022.3229161, 2022.

Gyawali, D. R. and Bárdossy, A.: Development and parameter estimation of snowmelt models using spatial snow-cover observations from MODIS, Hydrol. Earth Syst. Sci., 26, 3055 – 3077, https://doi.org/10.5194/hess-26-3055-2022, 2022.

Horn, J. E. and Schulz, K.: Spatial extrapolation of light use efficiency model parameters to predict gross primary production, J. Adv. Model. Earth Syst., 3, https://doi.org/https://doi.org/10.1029/2011MS000070, 2011.

Kim, M. and Kim, J.: Extending the coverage area of regional ionosphere maps using a support vector machine algorithm, Ann. Geophys., 37, 77–87, https://doi.org/10.5194/angeo-37-77-2019, 2019.

Krinner, G., Derksen, C., Essery, R., Flanner, M., Hagemann, S., Clark, M., Hall, A., Rott, H., Brutel-Vuilmet, C., Kim, H., Ménard, C. B., Mudryk, L., Thackeray, C., Wang, L., Arduini, G., Balsamo, G., Bartlett, P., Boike, J., Boone, A., Chéruy, F., Colin, J., Cuntz, M., Dai, Y., Decharme, B., Derry, J., Ducharne, A., Dutra, E., Fang, X., Fierz, C., Ghattas, J., Gusev, Y., Haverd, V., Kontu, A., Lafaysse, M., Law, R., Lawrence, D., Li, W., Marke, T., Marks, D., Ménégoz, M., Nasonova, O., Nitta, T., Niwano, M., Pomeroy, J., Raleigh, M. S., Schaedler, G., Semenov, V., Smirnova, T. G., Stacke, T., Strasser, U., Svenson, S., Turkov, D., Wang, T., Wever, N., Yuan, H., Zhou, W., and Zhu, D.: ESM-SnowMIP: assessing snow models and quantifying snow-related climate feedbacks, Geosci. Model Dev., 11, 5027–5049, https://doi.org/10.5194/gmd-11-5027-2018, 2018.

Di Marco, N., Avesani, D., Righetti, M., Zaramella, M., Majone, B., and Borga, M.: Reducing hydrological modelling uncertainty by using MODIS snow cover data and a topography-based distribution function snowmelt model, J. Hydrol., 599, 126020, https://doi.org/https://doi.org/10.1016/j.jhydrol.2021.126020, 2021.

N. Moriasi, D., G. Arnold, J., W. Van Liew, M., L. Bingner, R., D. Harmel, R., and L. Veith, T.: Model Evaluation Guidelines for Systematic Quantification of Accuracy in Watershed Simulations, Trans. ASABE, 50, 885–900, https://doi.org/https://doi.org/10.13031/2013.23153, 2007.

Parajka, J. and Blöschl, G.: The value of MODIS snow cover data in validating and calibrating conceptual hydrologic models, J. Hydrol., 358, 240–258, https://doi.org/10.1016/j.jhydrol.2008.06.006, 2008.

Scalzitti, J., Strong, C., and Kochanski, A. K.: A 26 year high-resolution dynamical downscaling over the Wasatch Mountains: Synoptic effects on winter precipitation performance, J. Geophys. Res. Atmos., 121, 3224–3240, https://doi.org/https://doi.org/10.1002/2015JD024497, 2016.

Tong, R., Parajka, J., Széles, B., Greimeister-Pfeil, I., Vreugdenhil, M., Komma, J., Valent, P., and Blöschl, G.: The value of satellite soil moisture and snow cover data for the transfer of hydrological model parameters to ungauged sites, Hydrol. Earth Syst. Sci., 26, 1779 – 1799, https://doi.org/10.5194/hess-26-1779-2022, 2022.

USDA: Chapter 2 Data Parameters, in: Snow Survey and Water Supply Forecasting National Engineering Handbook, 2010.

USDA: Chapter 6 Data Management, in: Snow Survey and Water Supply Forecasting National Engineering Handbook, 2014.

---

## Author Comment (AC2)

Dear Reviewer,

Thank you much for your review and valuable comments. Below you will find your referee comments (in black) and our responses (in blue).

With regards,

Atabek Umirbekov, on behalf of all authors

The paper addresses an important and compelling topic: the issue of choosing an adequate snow modelling scheme in the context of scarce data availability. This topic is particularly relevant for many areas of the world where instrumentation and monitoring is rather poor, yet the population depends on meltwater resources. The authors presented a machine learning-based model that requires simple and/or commonly available input data and no calibration. The model showed good performances in reproducing SWE both in the subset of stations not used for calibration and in two other remote, orographically complex and scarcely monitored stations. The model structure, training, validation and limitations are well explained and clear. The validation is extensive and considers point-wise and large-scale cases.

My suggestion is a major review. The motivations are the following. Generally, throughout the paper, I often found the literature review either insufficient or even absent. The description of the data used is scattered throughout the text, which doesn't help clarity. Figures often lack axes ticks, labels and/or units.

We are grateful for your comprehensive feedback on the manuscript and the valuable critical comments you provided. In response, we will enhance the literature review and expand the discussion of the important aspects that you have highlighted in your comments both here and below. We agree with your observation that the current version of the manuscript presents a mixing of data and methods, and we are committed to reorganizing them for clarity. Additionally, we will redesign incomplete figures and improve their overall organization, as you've suggested in your comments.

The comments are the following:

---                                    *MANUSCRIPT*                                    ---

**0. General comments:**

0.1 I suggest adding a comprehensive "Data" section where the authors can (a) list all the data they used, separating them in subsections for model training and validation, point-wise and large-scale; (b) roughly describe the geography/orography/data availability for the datasets they chose.

As requested, we will gather information on data used for both model training and evaluation under a separate sub-section "Data", and provide brief details on climate and topographical characteristics. The repositories indicated in the Data availability section will be updated with data and script used for large-scale evaluation of the model.

0.2 I suggest restructuring the final part of the paper with a freestanding "Model limitations" section and a "Conclusions" section encompassing and enhancing what is now in section "Summary".

As requested, we will separate 'Model limitations' into standalone section, and add "Conclusions" section to the manuscript

0.3 I suggest a re-reading and improvement of the English language, there are syntax/grammar errors in the text and the structure of some sentences is confusing (see comments for each section). Please check that the used tense is consistent along a section or paragraph.

0.4 Notations: throughout the text, figures and tables, please make the Celsius degree symbol consistent (°C); correct the Elevation unit from m to m a.s.l.; when a quantity is non-dimensional (i.e. NSE), please use the non-dimensional unit ([-]).

We will edit and improve clarity of the text across those highlighted sentences, and bring unit notations into consistency

**1. Introduction**

I suggest rewriting the Introduction by significantly expanding the state of the art and literary research, taking into account the following comments:

- L30: Suggested citation: Beniston M. (2008), Extreme climatic events and their impacts: Examples from the swiss alps. In: Díaz HFRJ (ed) Murnane, climate extremes and society. Cambridge University Press. New York. USA. pp: 147-164.

  Thank you for suggesting an appropriate reference for this sentence. We will refer to Beniston, 2008 in this line

- L31-39: This paragraph generally lacks references and examples on both kind of models; I suggest providing a small literature review.

  Thank you for this suggestion. We will provide a more detailed review of both types of the models and add supporting references (such as Essery, 2020; Jonas et al., 2019) .

- L37: "… research often opt for relatively simpler conceptual TI models…" references and examples are needed.

  We suggest to refer to Hock, 2003 and Ohmura, 2001 for this sentence.

- L40-41: I find this sentence too general and poorly supported by literature (the authors only provide one example). For example, in this recent study https://doi.org/10.5194/hess-26-3447-2022 the authors showed how a PB snow-hydrological model substantially outperformed a conceptual TI model. Both models were applied on the same spatial domain (catchment Dischma), and the TI model completely missed the snowmelt-induced discharge timing (see Figure 7 d-e).

  Thank you for pointing at the issue of insufficient references. We suggest to amend the sentence and supplement it with the following references: *"both types of models can produce similar results when calibrated for the current climate and applied to the same spatial domains (Bavera et al., 2014; Magnusson et al., 2011; Shakoor et al., 2018)."*
  In addition, we will add a new sentence into the paragraph: *"Models calibrated to the same conditions in the current climate can produce different predictions under climate change (Carletti et al., 2022)."*

- L51-60: I find this paragraph dedicated to the state of the art preceding the authors' work too short and general. I suggest expanding this section by better detailing the findings of previous works (upon which the authors rely for their work) and the critical issues of the previous works (which the authors seek to address in this paper).

  We will expand overview of machine learning applications for snow modelling as requested.

**2. Model description**

- The default threshold temperature value for rain/snow separation is set to -1 °C. Here, it would be necessary to justify this choice, or at least provide references, because this tuning parameter can vary a lot in snow/hydrological modelling (see for example https://doi.org/10.3390/cli9010008 for a TI model and https://doi.org/10.5194/hess-26-1063-2022 for a PB model).

  Thank you for this suggestion. We will add additional description with regard to Ts threshold, such as the following:

  *"The Ts constraint differs from classical temperature-based partitioning methods where the threshold defines precipitation in a binary way, as either 100% rainfall or 100% snow. In this model, snow-precipitation partitioning is simulated using its inherent learned relationships, but only until the temperature drops below Ts. At that point, any precipitation is regarded as 100% snow. For example, when the average temperature (TAVG) is 0°C, depending on other climatic and topographic variables, the model is likely to simulate a significant portion of precipitation as snow (around 75% of precipitation), even if Ts is set at -1°C."*

- L82-85: *"… and is available as a set of functions […] respectively"* If the subject is "a set of functions", then verbs should be "calculate" and "generate". Otherwise, the sentence as it is is unclear and I suggest rephrasing, dividing or better explaining.

  Indeed. We will correct the sentence accordingly.

- L110: *"As it was noted above, the SVR model has two tunable parameters: cost and gamma…"* Actually, gamma is never mentioned. The authors mention "sigma" on L99. Please clarify.

  We apologize for this confusion. We meant the same parameter, 'gamma', which is sometimes referred in literature as 'sigma'. We will stick to term 'gamma' throughout the text

**3. Model validation**

- L160: Please cite  https://doi.org/10.1016/0022-1694(70)90255-6

  Thank you for suggesting the reference. We will make a reference to Nash and Sutcliffe 1970 in line 160

- L180: As mentioned in Comment 0.1, Mendoza and Western Pamir are not mentioned earlier in the text as data used for validation and are only introduced here.

  Mendoza Andes and Western Pamir will be introduced in a new section 'Data'

- L199-200: Do the authors refer to Figure 4? If so, Figure 4 needs to be mentioned. See the comments about Figures.

We appreciate this suggestion. We will ensure that all figures are appropriately introduced and referenced in the mansucript.

- L202: *"… the rain-to-snow transition modelled using the metadata of the 520 validation SNOTEL stations."* Do the authors mean that there are observations/data on the transition between rain and snow for all the 520 stations? And how was that used in modelling? Please clarify.

   The main motivation behind this analysis is to have an understanding how the model simulates precipitation-snow partitioning during snow accumulation phase. The following new exert will provide additional details in this regard: *"Since the SNOTEL observations do not contain explicit information on precipitation-snow transition, we decided to use a sample of the dataset to simulate the transition depending on climate inputs (temperature variables) and topographical characteristics (e.g. Elevation). More specifically we have filtered the SNOTEL observations that closely fall on this phase by selecting observations that meet the following non-exhaustive main criteria: 1) observations for October or November when precipitation is non-zero 2) average temperature (TAVG) is less than 10 or higher than -10°C, 3) accumulated SWE is less than 20mm. We then run the model using the obtained sample of observations and estimated solid fraction of precipitation simulated by the model, i.e. amount of dSWE estimated by the model in respect to precipitation amount."*

- L206: *"… does not exceed 100%"* do the authors mean does not *reach* 100%?

   Yes, indeed, 'not reach 100%' is more appropriate and we will rephrase this part accordingly. Thank you for this correction.

- L210: I suggest justifying this sentence with a plot or a better explanation. Again, if this information is contained within some metadata, this needs to be explicitly stated.

   The histogram on the bottom left of Figure 7 could serve as supporting plot. We will make an appropriate cross-reference in the sentence.

- L241: How did the authors calibrate Ts? Please clarify.

   We calibrated Ts for each of the stations with the objective of maximizing the Nash-Sutcliffe Efficiency of the model`s simulations with respect to observed SWE. We will include this clarification into the text.

- L255-256: Can the authors verify this assumption? Shortly after, in the text, the authors write the same for the SnowMIP station SNB, so I assume it is possible?

We could use the example of the calibrated Ts values for the stations located in Sierra-Nevada as supporting evidence for this assumption; although some of the stations in this domain are located in close vicinity to each other, the calibrated Ts values exhibit high variability ranging from -1°C to -5°C). The main message of the sentence is that arbitrary altering Ts may lead to overcalibration through the error compensation effect. We will point to this issue in the Model limitations section.

- L292: The authors should explain the meaning of *"class balance accuracy"*.

  We will supplement this sentence with an explanation of class balance accuracy.

**4. Model sensitivity and uncertainty assessment**

- L305: Is there a reference for this method? If so, I suggest adding it.

  Yes, this method is explained in Fisher et al., 2018 and Greenwell et al., 2018. We will supplement these references into the sentence.

- L311: *"… depending on the phase considered …"* Do the authors mean "precipitation phase"? Please clarify. Also, the reference is missing.

  We refer to two general phases of snow metamorphosis: snow accumulation and snow ablation. We will make this clearer in the revised version of the manuscript. Our apologies for the missing reference; it was supposed to be a cross-reference to the Figure 10 further down.

- L316: What do the authors mean by *"relative comparison"*? Please clarify.

  In the given context, "relative comparison" means that the importance of those topographic variables is made in relation to other variables used by the model. We will rewrite this line in the text to make it clearer.

- L349: Please refer to Table 1 when addressing the different model settings.

  A cross-reference to the Table 1 will be included in the L348-349.

- L355: What do the authors mean by *"when outliers are controlled for"*? Please clarify.

  The boxplots in Figure 12 show extreme limits, which exclude outliers. More specifically, the minimum and maximum limits of the boxplots are determined by (1st Quartile - 1.5 * IQR) and (3rd Quartile + 1.5 * IQR), where IQR represents the interquartile range (Hu, 2020). To prevent confusion, we suggest that we remove the phrase '*when outliers are controlled for*' from the sentence.

**5. Summary**

- L375: The concept of equifinality is only addressed at the end of the paper but it is never mentioned earlier. The most important papers on equifinality are not cited (see https://doi.org/10.1016/0022-1694(89)90101-7, https://doi.org/10.1016/0309-1708(93)90028-E, https://doi.org/10.1016/j.jhydrol.2005.07.007). If overcoming equifinality is one of the aims of the paper, this needs to be addressed in the Introduction and also in the discussion of the results. And additionally, how does the model improve equifinality? This needs to be explained and justified. The results shown in Figure 12, for example, seem contradictory to this sentence, because there the authors show that one can obtain similarly good model performances with different sets of parameters.

  Thank you for suggested references. We will introduce issue of equifinality in snow modelling in the introduction and expand its discussion in respective part of the manuscript.

  In this sentence we refer to the challenge of calibrating multiple parameters in hydrological and snow modelling. This challenge is particularly prominent in hydrological modelling, where even relatively simple snow modules require calibration of at least two parameters: the precipitation-snow threshold and the degree factor. Considering that there are many other parameters for different components of a hydrological model, it would be easy to end up with multiple combinations of optimal model parameters. On other hand while our model contains only one tuneable parameter (Ts), it shows generally plausible performance in diverse climatic and topographic conditions upon using the default value of Ts. We hypothesize that replacing the snow module with a model that is based on generalizable empirical relationships may help to reduce the equifinality issue, especially when employing conceptual hydrological modelling.

  Figure 12 shows performance of four GEMS models that differ in a number of required inputs, but contain only a single parameter (Ts) which can be adjusted. All four models' performances depicted in figure 12 were obtained by using the default value of the Ts (-1°C)

  L383-385: This sentence is not clear. What do the authors mean by *"instrumental"*?

  We will edit the sentence, by replacing '*instrumental*' with '*helpful*' or '*useful*'.  Here we meant that "*balance (in) complexity, data requirement, and transferability... could be helpful for operational monitoring and hydrological modelling in data scarce domains.*"

- L385: Similarly for the equifinality, the problem of finding empirical relations and parametrizations is never addressed before in the text. If this is one of the aims of the paper, it needs to be addressed in the Introduction accompanied by proper references (as parametrizations of different kinds are already widely used in snow/hydrological modelling).

Thank you for raising this. We now recognize that the statement in this sentence may have been too assertive and requires further verification. We will remove this sentence from the manuscript.

- Please consider mentioning the undercatch selection issue within the Model limitation section.

By filtering observations for precipitation undercatch, we assume that the evaluation dataset is comparatively free of this issue. However our selection algorithm also filtered records where inconsistencies between accumulated precipitation and SWE may be reasoned by wind-induced snow-drift. Disentangling these two phenomena is challenging without further research. The model cannot capture/simulate snow-drifts, we acknowledge this limitation in lines 263-265 and explicitly stated it in lines 407-408.

---                                                        *FIGURES*                                                        ---

**General comments:**

- When a figure is composed by different subplots, as it is often the case in this paper, something that enhances clarity very much is naming each subplot differently, for example with letters like (a), (b)... And then, throughout the text, referring to each subplot like Figure 5a, Figure 5b etc.

- I suggest improving the figure referencing generally and throughout the whole text: often the authors describe the results referring to specific subplots of a same Figure by only mentioning the general Figure once at the beginning of the paragraph. Referring to each specific subplot before introducing each finding highlighted by the subplot increases clarity significantly.

Thank you for these recommendations. We will review the organization of the figures accordingly, and ensure they are properly introduced and referenced in the text.

**Specific comments:**

- Figure 2: Axes ticks and labels (latitude, longitude) are missing, legend is missing.

- Figure 3: Axes labels are missing.

- Figure 6: Left plots: missing adimensional symbol for NSE ([-]), missing unit for snow meltout date error (days?), missing y-axis label. Right plots: Missing axes ticks and labels (latitude, longitude).

- Figure 7: Same as above.

- Figure 8: y-axis label and units are missing.

- Figure 11: "Latitude" is spelled wrong, missing units, missing y-axis ticks and labels.

  Thank you for pointing out at these deficiencies. We will correct and align the figures accordingly.

**References:**

Bavera, D., Bavay, M., Jonas, T., Lehning, M., & De Michele, C. (2014). A comparison between two statistical and a physically-based model in snow water equivalent mapping. *Advances in Water Resources*, *63*, 167–178. https://doi.org/https://doi.org/10.1016/j.advwatres.2013.11.011

Beniston, M. (2008). Extreme climatic events and their impacts: examples from the Swiss Alps. In H. F. Diaz & R. Murnane (Eds.), *Climate Extremes and Society* (pp. 147–164). Cambridge University Press. https://api.semanticscholar.org/CorpusID:129970616

Carletti, F., Michel, A., Casale, F., Burri, A., Bocchiola, D., Bavay, M., & Lehning, M. (2022). A comparison of hydrological models with different level of complexity in Alpine regions in the context of climate change. *Hydrology and Earth System Sciences*, *26*(13), 3447–3475. https://doi.org/10.5194/hess-26-3447-2022

Essery, R. (2020). *Understanding and getting started with physically based snowmelt models*. The International Association of Hydrological Sciences. https://iahs.info/uploads/Commissions/ICSIH/ICSIH Understanding physically based snowmelt models.pdf

Fisher, A., Rudin, C., & Dominici, F. (2018). *Model Class Reliance: Variable Importance Measures for any Machine Learning Model Class, from the "Rashomon" Perspective*. https://api.semanticscholar.org/CorpusID:126316805

Greenwell, B. M., Boehmke, B. C., & McCarthy, A. J. (2018). *A Simple and Effective Model-Based Variable Importance Measure*. 1–27. http://arxiv.org/abs/1805.04755

Hock, R. (2003). Temperature index melt modelling in mountain areas. *Journal of Hydrology*, *282*(1), 104–115. https://doi.org/https://doi.org/10.1016/S0022-1694(03)00257-9

Hu, K. (2020). Become Competent within One Day in Generating Boxplots and Violin Plots for a Novice without Prior R Experience. *Methods and Protocols*, *3*(4). https://doi.org/10.3390/mps3040064

Jonas, T., Mcphee, J., Skiles, M., & Marks, D. (2019). *Understanding strengths and limitations of temperature-index snowmelt models* (Issue October). The International Association of Hydrological Sciences. https://iahs.info/uploads/Commissions/ICSIH/ICSIH snow modeling article FINAL.pdf

Magnusson, J., Farinotti, D., Jonas, T., & Bavay, M. (2011). Quantitative evaluation of different hydrological modelling approaches in a partly glacierized Swiss watershed.

*Hydrological Processes*, *25*(13), 2071–2084.
https://doi.org/https://doi.org/10.1002/hyp.7958

Ohmura, A. (2001). Physical Basis for the Temperature-Based Melt-Index Method. *Journal of Applied Meteorology*, *40*(4), 753–761. https://doi.org/https://doi.org/10.1175/1520-0450(2001)040<0753:PBFTTB>2.0.CO;2

Shakoor, A., Burri, A., Bavay, M., Ejaz, N., Ghumman, A. R., Comola, F., & Lehning, M. (2018). Hydrological response of two high altitude Swiss catchments to energy balance and temperature index melt schemes. *Polar Science*, *17*, 1–12. https://doi.org/https://doi.org/10.1016/j.polar.2018.06.007

---

## Author Response (AR2)

Dear Reviewers,

We would like to thank you once again for your review and helpful comments. We believe they helped us to substantially improve the manuscript both content- and structure-wise. Below you will find your referee comments (in black) and our responses (in blue).

With regards,

Atabek Umirbekov, on behalf of all authors

**RC1**: **Matthieu Lafaysse**

**General comments**

Umirbekov et al. present a new machine learning approach to simulate snow mass with parcimonious data input and an extremely low numerical cost. The evaluation framework is really interesting as it includes independent data removed from the calibration dataset, but also the state-of-the-art ESM-SnowMIP dataset including challenging climate and environment conditions beyond those of the calibration dataset, and finally a spatialized application with more uncertain forcing data and evaluation data derived from remote sensing. Of course, the potential of machine learning has to be considered in snow modelling and I think this paper can be a significant contribution on that topic. The results clearly challenge physical models, even if obviously the output variables are not sufficient for all applications.

Nevertheless, I think the description of methods and results is sometimes a bit too fast in the current version of the manuscript and that some details are missing for an accurate understanding and interpretation of results. In general, figures are not really introduced in the main text. I would also have expected more in-depth discussion of the advantages and disadvantages of this approach compared to physical approaches and other machine learning approaches in the light of presented results and previous literature, and also discussions about the possibility to disentangle errors due to the forcing and to the algorithm itself. Maybe, the chosen structure of the paper that mixes results description and results discussion is partly responsible for this sometimes incomplete discussion. Finally the choice to try to recalibrate the Ts parameter is sometimes confusing especially when it's done on evalution datasets, as it leads to unrealistic values and overcalibration.

I also have some specific comments or questions below that can probably be addressed rather easily by the authors during the revision process.

Dear Dr. Matthieu Lafaysse,

We appreciate your comprehensive feedback on the manuscript and are grateful for your valuable comments. In response to your suggestions, we had incorporated a comparison with simulations from the ESM-SnowMIP study (Krinner et al., 2018), and included a brief discussion of the advantages and disadvantages of the model in comparison with machine learning and physical snow models. We had introduced all figures in the main text for better context. We had restructured the manuscript which allows better orientation for readers through the text. We had also provided a more detailed description in the main text regarding the precipitation-snow partitioning and the 'Ts' parameter to prevent any confusion with traditional temperature-based partitioning methods.

**Detailed comments**

Section 2.1 The choice of SVR relatively to other machine learning algorithms is not discussed. I would suggest to add a quick summary of advantages and disadvantages compared to the most classical algorithms available in literature (random forests, convolutional neural network, simpler regressions, etc.)

Thank you for this suggestion. We had added into the introduction a brief overview of machine learning application to snowpack modeling (lines 61-70 in the tracked changes version):

> "In terms of ways in which machine learning (ML) has been integrated into snowpack modeling, the respective research studies can be grouped into several main approaches. One common approach is estimating spatial distribution of snowpack by applying ML-supported interpolation of sparse snow observations and using topographical features, meteorological and satellite data (Broxton et al., 2019; Mital et al., 2022). Other studies have explored potential of satellite radar data for direct detection of instantaneous properties of snowpack (Santi et al., 2022; Daudt et al., 2023). In cases where one or multiple gridded snow products are available, ML can be employed for a better prediction through assimilation of multiple estimates or bias-correction (Shao et al., 2022; King et al., 2020). A few recent studies managed to apply ML in a manner consistent with traditional snow models, explicitly modeling snow mass accumulation and melt dynamics (Vafakhah et al., 2022; Duan et al., 2023)."

In addition, we included a brief paragraph into the discussion section summarizing some our experiments and hypotheses (lines 555-577):

> "Machine learning is still an emerging sub-field in hydrological and snow modelling, with most existing applications predominantly limited to snowpack interpolation or the detection of its instantaneous state through the assimilation of ground-truth and active satellite radar data. In contrast, GEMS provides a modelling framework similar to traditional snow modelling approaches, by simulating snowpack in a dynamic manner and leveraging climate and topographic inputs commonly used in snow models. Moreover, the revealed variable importance aligns with the general physics governing how climate affects changes in snowpack during its accumulation and ablation

phases. Some recent studies employing machine learning algorithms (Vafakhah et al., 2022; Duan et al., 2023) also simulate snowpack in a dynamic manner and demonstrate promising results. It remains unclear however what the extrapolation capacity is of those models beyond the spatial domain where they had been trained on, which is a known challenge in machine learning. Transferability of the models can be also constrained by the use of complex climate inputs or specific local datasets that may not be available elsewhere. From these perspectives, GEMS offers a higher degree of parsimony in terms of required input variables and, more importantly, a proven ability to generalize outside of the training domain.

We have evaluated several other data-driven techniques for the model development, including multivariate linear regression, Gaussian process, Random Forests, and Gradient Boosting Machines (not shown here). When evaluating on the training dataset, the performance of most models was either lower or equivalent to SVR; however, even in the latter case their accuracy on the evaluation dataset was worse. Experiments in other fields indicate that SVR has relatively better extrapolation potential on unseen data (Horn and Schulz, 2011; Kim and Kim, 2019), which may explain why it outperformed other algorithms. We haven't examined neural network algorithms since they take more computer resources during training, and evidence suggests that they tend to underperform relative to other machine learning ML techniques when applied to tabular data (Borisov et al., 2022; Shwartz-Ziv and Armon, 2022). To make definitive judgments with regard to performances of different machine learning algorithms, however, would require a more extensive intercomparison experiment that is outside the scope of this paper.

Can you define more explicitely i, j, N, xi, xj, X ?

We added notations for the variables and parameters denoted in the SVR formula (lines 116-126)

I understand from Fig.1 and Eq. 2 that when temperature is below the -1°C threshold and precipitation is zero, then dSWE is always equal to 0. Is that correct ? How often does this assumption fail in the training or evaluation dataset? Does this imply an intrinsic limitation of GEMS for transferability on steep slopes where the surface energy balance can be positive even at negative temperatures ? (I think it does.)

Thank you for these guiding questions. We recognize this as one of the model constraints and noted it in the Model limitations section (lines 502-505):

"When temperature is below the -1°C Ts threshold and precipitation is zero, GEMS will automatically estimate daily change in SWE as 0 mm. The model thus does not account for snow sublimation process, which can occur even when temperatures are below freezing. This differs from some other snow models, particularly those based on energy balance, which can estimate snow sublimation."

Section 2.2

The authors say they « fine-tuned the hyperparameters so that the model produces similar levels of accuracy when applied to observations from the same stations for 2019 and

2020. » I understand the general idea but the detailed procedure is not accurately described. Can you describe the detailed protocol for this « fine-tuning » ?

We complemented respective sub-section with the following details (lines 130-134):

> "The hyperparameter calibration process involved an exhaustive 'grid-search' technique, which systematically explored all possible combinations within predefined parameter ranges. Ultimately, we selected the hyperparameter configurations that resulted in the lowest root mean squared error between simulated and observed dSWE during both model training on observations from 2017-2018 and its testing on observations from 2019 and 2020."

As solid precipitation measurements are prone to large measurement errors and is one of the main predictor of the model, I would have expected more details about precipitation gauges used in the SNOTEL network, procedures applied to account for undercatch, and if possible estimated uncertainties.

We added the following details into the Data section (lines 165-168):

> "SNOTEL precipitation gauges may also be susceptible to solid precipitation undercatch, especially when snowfall occurs in windy conditions (USDA, 2014). Scalzitti et al., 2016 provide a comprehensive review of the issues associated with precipitation undercatch, highlighting reported undercacth ranging from 11% for snowfall under 2m/sec wind speed to more than 30% during intense snowstorm events."

Section 3 I think « Model evaluation » would be a more appropriate title than « model validation » as a model can never be considered as fully validated.

Thank you for this suggestion. We changed the section title to "Model evaluation".

The authors say « we excluded stations that exhibit precipitation undercatch, which we formulate as when SWE accumulated by March is greater than the accumulated precipitation during October to March. ». I would expect all stations to be affected by precipition undercatch and total SWE to be always higher than raw precipitation measurements. Do you apply a specific threshold to only eliminate major undercatch ? Or do you use precipitation timeseries that are already corrected for precipitation undercatch following WMO recommendations ? My misunderstanding is probably linked to the lack of details in Section 2.2 as previously mentioned.

Then, was this selection procedure also apply to the training dataset ? If not, why ?

We appreciate these comments and questions. We agree that this part needs more clarifications, and included some additional explanatory exerts, which include:

line 168-171

"To ensure data accuracy, we cleaned the training dataset by removing observations with inconsistencies between daily precipitation and snow mass accumulation. These inconsistencies refer to cases when the daily increase in SWE exceeded the reported daily precipitation."

The selection approach differed for the evaluation dataset because we aimed to retain as many stations as possible for evaluation and besides that the model requires complete daily time series without missing observations.

Line 189-192

"This approach enabled us to include more stations in the evaluation dataset while excluding only those hydrological years that exhibited inconsistencies between these variables. When using this filter, we did not set any specific threshold for the magnitude of inconsistencies, nor did we make corrections to the precipitation time series."

It should be also noted that we included a criterion that required at least five hydrological years of observations for a station to be part of the evaluation dataset. Consequently, some SNOTEL stations were excluded based on this specific requirement.

Section 3.1

L193 I would suggest to start by a sentence presenting the Figure before providing its interpretation.

We have introduced the Figure 4 in the text before its interpretation (line 241-242)

In Figure 4 « actual » should be replaced by « observed ». Is there a reason to present the simulations in the X axis and not in the Y axis (that would be more common for a scatter plot) ?

Thank you for pointing at this. We have replaced 'actual" with "observed" and also modified X and Y axis accordingly (Figure 4, line 248)

In Figure 5, it is not immediate to understand what is represented because the caption is not self-sufficient and the description in the text is also too vague. The definition of TAVG should be remind in the caption. Then what does represent a single point ? A station and a date ? Then, this solid fraction of precipitation does not really appear in model description, neither in Figure 1 neither in the Equations, so it is difficult to understand how this diagnostic is obtained from the provided model description. The reason for providing this Figure is also unclear as finally these outputs are not really used as a fixed temperature threshold finally replaces the values obtained by the algorithm. This needs to be clarified.

We appreciate your suggestion to incorporate additional clarifications into this section. We added more description into the text (line 255-262):

"Since the SNOTEL observations do not contain explicit information on precipitation-snow transition, we decided to use a sample of the dataset to simulate the transition depending on climate inputs (temperature variables) and topographical characteristics (e.g. elevation). More specifically we have filtered the SNOTEL observations that closely fall on this phase by selecting observations that meet the following non-exhaustive main criteria: 1) observations for October or November when precipitation is non-zero 2) average temperature (TAVG) is less than 10 or higher than -10°C, 3) accumulated SWE is less than 20mm. We then run the model using the obtained sample of observations and estimated solid fraction of precipitation simulated by the model, i.e. amount of dSWE estimated by the model in respect to precipitation amount. **Error! Reference source not found.** depicts the rain-to-snow transition modelled using the metadata of the 520 validation SNOTEL stations."

As for the other Figures, introducing quickly Figure 6 would be helpful before providing the results analysis. In the description of the results of Figure 6, detailed references to the subplots would help to follow results description.

We introduced the Figure 6 in the text, lines 290-291.

Isn't the maxSWE score more representative of the quality of input precipitation than of the skill of the SVR model ?

Yes, given the temperature threshold, we assume that maxSWE might be more representative of precipitation input accuracy. However, since a portion of the simulated maxSWE is influenced by the model's simulation of dSWE (at temperatures above the $T_S$ threshold), we think it is reasonable to keep maxSWE as one of the metrics.

L252-254 If removing stations with incorrect measurements is understandable, removing stations with snow drift should be avoided as snow drift is not a measurement error, it's a natural process challenging to reproduce with physical models and also maybe with machine learning models, but the general ability or unability of any model to reproduce snow conditions should account for places where snow drift happen.

Thank you for raising this concern. Instances where recorded maxSWE exceeds accumulated precipitation may be due to snow-drift, precipitation undercatch, or a combination of both factors. Unfortunately, attributing these inconsistencies to individual factors may require a separate research effort. We therefore had to exclude those stations from the evaluation, but we note the inability of the model to capture snow-drift in line 323-325 and explicitly state this as a model`s limitation in line 498-500.

L255-256 You mean that an overcalibration is obtained due to error compensation between snow drift and rain-snow transition ? Could the sentence be more clear ?

Thank you. Yes indeed, we meant that overcalibration may lead to error compensation. However, we now realize that beside the snow drifting, there might be several other

contributing factors leading to overestimation of SWE, such as sublimation, effect of dense canopy, and rain-on-snow events. Unfortunately, we can not delineate/ verify these factors within the scope of this manuscript, but we believe they should be noted since they also define limitations of model. Therefore, we rewrote this passage in the following manner (lines 320-326):

> "While the median of the adjusted TS values for all stations agrees with its default threshold (-1 °C), the density distribution also shows a high frequency of calibrated Ts resulting at the lowest bound of -5 °C Figure 7f). This suggests that, in many cases where calibrated Ts values approach the lowest boundary, the model simulations might have been overcalibrated, resulting in error compensation. The overestimation of SWE at these locations can be attributed to several factors that the model does not account for, including effect of dense vegetation, wind induced snow-drift, sublimation, and rain-on-snow events which may be frequent phenomena in the mountain areas (Li et al., 2019; Boniface et al., 2015; Kirchner et al., 2014; Sexstone et al., 2018)."

Section 3.3

Again, an introduction of Figure 8 in the text would be useful.

We have introduced the Figure 8 in the text (lines 326-327).

L266-267 It is not obvious which value of NSE should be considered as « acceptable ». Indeed, NSE is easily high when dealing with variables with a high seasonal cycle. What would be the NSE value of the daily interannual mean of observed SWE ? Is the 0.7 value at Sapporo better than such a reference score ?

We intended to refer to some categorizations of NSE across multiple studies (e.g. N. Moriasi et al., 2007). However, we recognize that these classifications, designed for hydrological models, might not be directly applicable for classifying snow model outputs. Therefore, we revised sentences with qualitative classifications in the text like this one.

L269-270 This could be moved to the Method section

Here we refer to the limitation with one of the input variable for the SOD station, which may be a potential source of uncertainty for the model simulations. We therefore assume that it is more appropriate to keep this exert in the same paragraph. However, we slightly modified respective lines to make ours message clearer (lines 347-351):

> "It is important to note that in terms of latitude and thus the range of daylengths, the SOD station is situated much beyond the range of the data utilized to pre-train the GEMS model. In addition, since the Global Continuous Heat-Insolation Load Index (CHILI) does not extend beyond the arctic circle. To estimate it for SOD, we used the nearest known value and assuming flat terrain, though our estimate may have some uncertainty."

L274-280 As it was already noticed with the SNOTEL dataset that local calibration of the Ts threshold leads to severe error compensations, and as the purpose of the application of the GEMS system on the ESM-SnowMIP dataset is to assess its spatial transferability beyond its training dataset, I am not really convinced of the interest to test again to recalibrate locally this threshold on each ESM-SnowMIP site. The conclusions that again this leads to overcalibration and errors compensations were rather expected, so I would suggest to remove thisanalysis.

Thank you for these insights and the suggestion. We acknowledge that calibrating the Ts threshold may result in error compensations. However, the results do not provide insights into the extent of these compensations. As previously described, the Ts threshold in the model differs from the classical temperature-based threshold method. For instance, when Ts is set at -3°C and temperature (TAVG) is 0°C, the model will likely classify a larger portion of precipitation as snow (Figure 5). Nevertheless, we recognize that calibration in general might be inappropriate when assessing the spatial transferability of the model. Hence, we removed the calibration analysis for Snow-MIP stations from the manuscript.

Apart from model evaluation, calibration could still be useful during model application, particularly when local precipitation-snow partitioning patterns are known. In light of this, we more explicitly acknowledged the risk for error compensation due to calibration in the model limitations section, and there recommended calibration only if local precipitation-snow partitioning patterns are known (lines 505-507):

> "Furthermore, the evaluation on the SNOTEL dataset suggests that significant adjustments of the $T_S$ threshold imposes a risk of error compensation due to over-calibration. Therefore, we recommend adhering to the default value of $T_S$ (-1°C), unless local precipitation-snow partitioning patterns are well understood."

Section 3.4

Again an introduction of Figure 9 is missing.

We introduced Figure 9 in the text (line 382).

My feeling is that the level of discussion in this section is not as advanced as for the evaluation on ESM-SnowMIP sites. How does this skill in terms of snow cover extent compare with physical models ?

Thank you for this suggestion. We acknowledge that this section's content is not as comprehensive as other sections, particularly in terms of comparison with the performance of physical models. However, the extensive computational burden for such a comparison present a significant challenge to us. Instead we added some clarifications of why we conducted this analysis (lines 383-388):

> "The primary objective of this analysis was to test and demonstrate the model's transferability to regions with complex terrain and lacking in-situ SWE data. We assume that if the extent of the simulated SWE aligns well with the remote-sensed snow cover, then the simulated SWE is likely to contain less uncertainty. This assumption is also based on fact that remote sensed snow cover is increasingly used for parameter calibration or uncertainty reduction in snow modules of hydrological models (e.g. Parajka and Blöschl, 2008; Gyawali and Bárdossy, 2022; Tong et al., 2022; Di Marco et al., 2021)."

Section 4.1

L312 Reference error.

We apologize for this error, the missing part was intended as a reference to Figure 10. We corrected this in a new version of the manuscript and appropriately introduced Figure 10 in the text (lines 411-412).

L315 Could the relatively low contribution of the heat-insolation index be possibly explained by an unsufficient variability of this predictor in the training dataset ?

Yes, this is what we intended to state. We revised the sentence making this message clearer (lines 421-424).

In mountainous areas, shadows and slope inclinations are a major factor to explain melting. But I assume that all observations correspond to flat areas, and maybe the variability of shadows in the SNOTEL network is neither representative of the variability of topographic conditions in mountains. This is important to discuss as it could limit the possibility to apply this algorithm on areas with complex topography.

We appreciate these comments and suggestions. Indeed, the SNOTEL stations utilize flatbed pillows, but are primarily situated in mountainous regions. However, the introduction of heat-insolation index (CHILI) helps to capture effects and variability of terrain-induced shadowing. Despite of this, we have introduced the following passage into the limitations section (lines 508-512):

> "As discussed in the Section 4.2and illustrated in Figure 11a, the model demonstrates relatively better performance in mountainous areas compared to lower elevations. However, the training dataset used to elaborate the model may be less representative of locations with lowest CHILI indices (Figure 2d). Lower CHILI indices often correspond to sites significantly shadowed by terrain or situated at higher latitudes or both. This discrepancy may be an additional source of model uncertainty. "

Section 4.2

I am wondering how much this conclusion is affected by the choice of NSE to quantify errors. Indeed, as this score is highly influenced by the existence of a seasonal cycle, it is rather normal to get better scores with deeper snowpacks that exhibit a very strong seasonality than on sites with more intermittent snow cover. Considering other scores (for instance a Root Mean Square Error), I would not be surprised that stations with the poorest performance would be reversed. Can you comment on that topic ?

Thank you for your guiding questions. We agree that NSE alone may not adequately distinguish between cases of 'good' and 'poor' model performance, and use of different metrics would likely result in varying compositions of these two performance groups. We tested normalized RMSE as a metrics but got similar results in terms of 'poor' and 'good' simulations across the stations. We therefore haven't amended this analysis much, except a minor modification: instead of using a threshold to delineate between the 'poor' and 'good' simulations (NSE less or greater than 0.7), we compared the lowest and highest quartiles of NSE across the stations (lines 430-432).

L375 The authors say that « GEMS also addresses the equifinality issue that is pertinent to hydrological and snow modelling. » but the only parameter they have introduced (Ts threshold) clearly raises a very strong equifinality resulting in possible overcalibration to compensate various possible errors including snow drift, precipitation undercatch, etc.

We assume that this sentence is now justified, considering the preceding explanation of how Ts works in the model, how it differs from temperature-based partitioning methods, as well as our intention to stick to the default Ts in our recommendations. In this sentence we refered to the challenge of calibrating multiple parameters in hydrological and snow modelling. This sentence is now a bit modified and expanded with the following clarification (line 537-543):

> "In addition to avoiding computationally demanding calibration, GEMS may also alleviate the challenge of equifinality of model parameters that is pertinent to modelling environmental systems. The issue of equifinality is particularly pronounced in hydrological modeling, where even relatively simple snow modules require the calibration of at least two parameters: the precipitation-snow threshold and the degree-melt coefficient. Considering that there are many other parameters for remaining components of a hydrological model, it would be easy to end up with multiple combinations of optimal parameters. In contrast GEMS model shows generally plausible performance in diverse climatic and topographic conditions upon using the default value of $T_S$."

L388 « GEMS can, for instance, provide information for the parameterization of physics-based models, e.g. precipitation phase partitioning and its elevational dependence ». I don't see how the results presented here suggest this conclusion and considering the strong risk of overcalibration of this Ts value (leading to clearly unrealistic values below -5°C), I am not convinced at this point that GEMS could help me to discriminate between snow and rain.

As mentioned earlier, we acknowledge that calibrating Ts poses a risk of error compensation, though considering how Ts operates in the model, the extent of overcalibration maybe not as pronounced as it would be with traditional temperature-based thresholds. Despite this, we recognize that the statement in this sentence may have been too assertive and requires further verification, therefore we removed this sentence from the manuscript.

There is a section 5.1 but not any section 5.2. Maybe a subtitle for the first part of Section 5 is missing.

As it was also recommended by Reviewer 2, we deliniated the section into two separate sections in a new version of the manuscript: section 5 'Model Limitations' and section 6 'Summary'.

L393-400 The authors discuss the limitations of their approach relatively to forest areas but they seem to have intentionnally remove the 3 forest sites of the ESM-SnowMIP dataset from their evaluations. This should at least be discussed if there is a valid reason for that. But even if the model skill is lower on the 3 Canadian forest sites, I would have included these sites in the evaluations to provide concrete results to support this discussion.

Indeed, we haven't evaluated the model on the three Canadian sites but because at that time we couldn't precisely locate the sites to determine CHILI parameters. We have included these sites for the model evaluation in the next version of the manuscript (lines 354-357):

> "The performance of the model exhibited notable disparities across three forested locations in Canada (OAS, OBS, OJP). In comparison to other sites, the model's performance at these sites was relatively inferior, indicated by NSE values ranging between 0.44 and 0.66 and maxSWE errors spanning from 15% to 30%. This observation suggests a diminished performance of the model in environments characterized by dense canopy interception. "

L408-410 Unfortunately, blowing snow can be an important process even at large scale especially in polar regions. So large scale applications of the system may still be affected by this limitation.

We removed that part of the sentence.

The discussion do not compare the skill of this approach with the skill of physical models while similar metrics are provided at the same sites in Ménard et al., 2021, and other

evaluations are also available in the literature for snow cover extent. I think this would be important to consider as well.

We appreciate this suggestion. We have compared the skill of the model in terms of NSE with that of physical models that participated in ESM-SnowMIP, using model simulations presented in Krinner et al., 2018. It should be noted however that this comparison has some limitations, since participants of the ESM-SnowMIP study didn't have possibility to adjust model parameters, rain-snow transition in particular. Respective new exerts include the following:

Lines 358-364

"For reference,Table 4 also provides the NSE of simulations produced by models that participated in ESM-SnowMIP. With the exception of the SNB site, ESM-SnowMIP simulations had lower NSE than those of GEMS simulations. However, a direct comparison between GEMS and ESM-SnowMIP simulations is not possible because evaluation data were not provided to the ESM-SnowMIP participants in advance and rain-snow transitions were prescribed in the driving data (Ménard et al., 2019). ESM-SnowMIP participants thus had no opportunity to enhance model performance by adjusting parameters.

Lines 532-536 in the Summary section:

"The model evaluation suggests that GEMS achieves comparable performance to physical snow models, as evidenced by comparing with simulations from ESM-SnowMIP. A more appropriate comparison might necessitate adjustment of physical model parameters, which was not investigated in ESM-SnowMIP. Nevertheless, the evaluation outcomes allow us to conclude that, at the very least, GEMS with its default TS parameter exhibits superior spatial transferability compared to physical models with unadjusted parameters."

The discussion or final summary also lack comments about the strengths and weaknesses of their results compared to the literature cited in the introduction applying machine learning to predict snow mass.

We have added our perspective on the strengths and weaknesses of our model approach compared to other cases of snow models utilizing machine learning (lines 555-577):

"Machine learning is gaining more space in snow modelling, with existing applications predominantly focusing on snowpack interpolation or the detection of its instantaneous state through the assimilation of ground-truth and active satellite radar data. GEMS provides a modelling framework similar to traditional snow modelling approaches, by simulating snowpack in a temporally progressive manner and leveraging climate and topographic inputs commonly used in snow models. Moreover, the revealed variable importance aligns with the general physics governing how climate affects changes in snowpack during its accumulation and ablation phases. Some recent studies employing machine learning methods (Vafakhah et al., 2022; Duan et al., 2023) also simulate snowpack in a temporal manner and demonstrate robust performance, though spatial extrapolation limits of those algorithms remain unclear. Another recent study (Wang et al., 2022) presents

promising results for a deep learning-based approach, showcasing its superior spatial transferability compared to enhanced temperature index model across the United States. Nevertheless, the applicability of these models beyond their targeted regions may be questionable due to dependance on climate inputs or locally-specific data that may not be available elsewhere. From these perspectives, GEMS offers a higher degree of parsimony in terms of required input variables and, more importantly, a proven ability to generalize outside of the training domain.

We have evaluated several other data-driven techniques for the model development, including multivariate linear regression, Gaussian process, Random Forests, and Gradient Boosting Machines (not shown here). When evaluating on the training dataset, the performance of most models was either lower or equivalent to SVR; however, even in the latter case their accuracy on the evaluation dataset was worse. Experiments in other fields indicate that SVR has relatively better extrapolation potential on unseen data (Horn and Schulz, 2011; Kim and Kim, 2019), which may explain why it outperformed other algorithms. We haven't examined neural network algorithms since they take more computer resources during training, and evidence suggests that they tend to underperform relative to other machine learning ML techniques when applied to tabular data (Borisov et al., 2022; Shwartz-Ziv and Armon, 2022). To make definitive judgments with regard to performances of different machine learning algorithms, however, would require a more extensive intercomparison experiment that is outside the scope of this paper."

Furthermore, the outputs of the model are currently limited to SWE while several snow-sensitive applications require more variables (e.g. surface temperature for NWP and climate modelling, snow internal properties for remote-sensing retrieval algorithms or avalanche forecasting). This limitation should also be mentioned with possibly discussions about the feasability to extend this approch to more variables.

Thank you for this suggestion. We have included this limitation and complement it by presenting our perspective on the snow processes to which our approach may be applicable (lines 544-550):

"One difference between GEMS and physics-based models lies in the number of outputs they generate. While GEMS is specifically designed for simulating only SWE, comprehensive physics-based snow models produce a broader spectrum of outputs that provide valuable insights into other snow properties. We assume that machine learning could become helpful in modelling some of these snow properties. For example, previous studies have shown how simple empirical models can effectively derive snow depth from SWE measurements and vice versa (Aschauer et al., 2023; Hill et al., 2019). We assume that a similar approach to GEMS could be scalable for estimating snow depth by incorporating additional variables such as snow age etc. "

**RC2: Anonymous Referee #2**

The paper addresses an important and compelling topic: the issue of choosing an adequate snow modelling scheme in the context of scarce data availability. This topic is particularly relevant for many areas of the world where instrumentation and monitoring is rather poor, yet the population depends on meltwater resources. The authors presented a machine learning-based model that requires simple and/or commonly available input data and no calibration. The model showed good performances in reproducing SWE both in the subset of stations not used for calibration and in two other remote, orographically complex and scarcely monitored stations. The model structure, training, validation and limitations are well explained and clear. The validation is extensive and considers point-wise and large-scale cases.

My suggestion is a major review. The motivations are the following. Generally, throughout the paper, I often found the literature review either insufficient or even absent. The description of the data used is scattered throughout the text, which doesn't help clarity. Figures often lack axes ticks, labels and/or units.

Dear Reviewer

We are grateful for your valuable feedback and comments. In response, we have enhanced the literature review and expanded the discussion of the important aspects that you have highlighted in your comments both here and below. We agree with your observation that the first version of the manuscript presents a mixing of data and methods, and we have reorganized them for clarity. Additionally, we redesigned incomplete figures and improved their overall organization, as you've suggested in your comments.

The comments are the following:

---                                    *MANUSCRIPT*                                    ---

**0. General comments:**

0.1 I suggest adding a comprehensive "Data" section where the authors can (a) list all the data they used, separating them in subsections for model training and validation, pointwise and large-scale; (b) roughly describe the geography/orography/data availability for the datasets they chose.

As requested, we have gathered information on data used for both model training and evaluation under a separate section "Data", and provided brief details on climate and topographical characteristics. Sources of all used data had been previously indicated in the Data availability section.

0.2 I suggest restructuring the final part of the paper with a freestanding "Model limitations" section and a "Conclusions" section encompassing and enhancing what is now in section "Summary".

As requested, we have separated 'Model limitations' into standalone section, and added "Summary and Conclusions" section to the manuscript.

0.3 I suggest a re-reading and improvement of the English language, there are syntax/grammar errors in the text and the structure of some sentences is confusing (see comments for each section). Please check that the used tense is consistent along a section or paragraph.

0.4 Notations: throughout the text, figures and tables, please make the Celsius degree symbol consistent (°C); correct the Elevation unit from m to m a.s.l.; when a quantity is non-dimensional (i.e. NSE), please use the non-dimensional unit ([-]).

We have edited some sentences across the text according to your comments per each section below, and corrected unit notations accordingly.

**1. Introduction**

I suggest rewriting the Introduction by significantly expanding the state of the art and literary research, taking into account the following comments:

- L30: Suggested citation: Beniston M. (2008), Extreme climatic events and their impacts: Examples from the swiss alps. In: Díaz HFRJ (ed) Murnane, climate extremes and society. Cambridge University Press. New York. USA. pp: 147-164.

  Thank you for suggesting an appropriate reference for this sentence. We added a reference to Beniston, 2008 in line 30

- L31-39: This paragraph generally lacks references and examples on both kind of models; I suggest providing a small literature review.

Thank you for this suggestion. We have added supporting references in line 39 (such as Essery, 2020; Link et al., 2019) that provide descriptions of two types of snow models, though we have not extended the text with particular model examples and their description. In our opinion adding model examples and their descriptions will require new extensive paragraphs which would divert a focus of the of the introduction.

- L37: "… research often opt for relatively simpler conceptual TI models…" references and examples are needed.

  We have added references to Hock, 2003 and Ohmura, 2001 for this sentence.

- L40-41: I find this sentence too general and poorly supported by literature (the authors only provide one example). For example, in this recent study https://doi.org/10.5194/hess-26-3447-2022 the authors showed how a PB snow-hydrological model substantially outperformed a conceptual TI model. Both models were applied on the same spatial domain (catchment Dischma), and the TI model completely missed the snowmelt-induced discharge timing (see Figure 7 d-e).

  Thank you for pointing at the issue of insufficient references. We have amended the sentence and supplemented it with the following references (lines 43-45):

  "Despite the differences in the number of internal processes represented and the corresponding data requirements, both types of models produce similar results when calibrated and applied to the same spatial domain and same climatic conditions (Kumar et al., 2013; Bavera et al., 2014; Magnusson et al., 2011; Shakoor et al., 2018).

  In addition, we added a new sentence into the paragraph (lines 48-50):

  "Models calibrated to the same conditions in the current climate can produce different predictions under climate change (Carletti et al., 2022)."

- L51-60: I find this paragraph dedicated to the state of the art preceding the authors' work too short and general. I suggest expanding this section by better detailing the findings of previous works (upon which the authors rely for their work) and the critical issues of the previous works (which the authors seek to address in this paper).

  We have expanded the overview of machine learning applications for snow modelling with the following passage (lines 61-70):

  "In terms of ways in which machine learning (ML) has been applied for snowpack modeling, the respective research studies can be grouped into several main approaches. One common approach is estimating spatial distribution of snowpack by applying ML-supported interpolation of sparse snow observations and using topographical features, meteorological and satellite data (Broxton et al., 2019; Mital et al., 2022). Other studies have explored potential of satellite radar data for direct detection of instantaneous properties of snowpack (Santi et al., 2022; Daudt et al., 2023). In cases where several gridded snow products are available, ML can be employed for a better prediction

through assimilation of multiple estimates or bias-correction (Shao et al., 2022; King et al., 2020). A few recent studies managed to apply ML in a manner consistent with traditional snow models, explicitly modeling snow mass accumulation and melt dynamics (Vafakhah et al., 2022; Duan et al., 2023; Wang et al., 2022). However, in many instances, most of the noted approaches also rely on is-situ observations or extensive set of regional reanalysis variables, which again restricts their wider applicability due to unavailability of such data in many regions. Furthermore, the ability of pretrained machine learning models to generalize to new geographic and climatic domains remains another challenge; machine learning models often perform less well outside the data distribution used to train them (Chase et al., 2022; Hernanz et al., 2022)."

**2. Model description**

- The default threshold temperature value for rain/snow separation is set to -1 °C. Here, it would be necessary to justify this choice, or at least provide references, because this tuning parameter can vary a lot in snow/hydrological modelling (see for example https://doi.org/10.3390/cli9010008 for a TI model and https://doi.org/10.5194/hess-26-1063-2022 for a PB model).

  Thank you for this suggestion. We have added additional description with regard to $T_S$ threshold, such as the following (lines 274-280):

  > "Here it is important to note that the TS constraint in the GEMS model differs from classical temperature-based partitioning methods where the threshold defines precipitation in a binary way as either 100% rainfall or 100% snow. The model simulates snow-precipitation partitioning only until the temperature drops below TS, at which point any precipitation is regarded as 100% snow. For example, when the average temperature (TAVG) is 0°C, using the assimilated statistical relationships the model will likely simulate some portion of precipitation as snowfall. As illustrated in the Figure 5, **Error! Reference source not found.** at TAVG around of 0°C, the model, on average, simulates around 75% of precipitation as snowfall. Depending on other input variables this ratio varied from approximately 25% to as high as 95%."

- L82-85: "*... and is available as a set of functions [...] respectively*" If the subject is "a set of functions", then verbs should be "calculate" and "generate". Otherwise, the sentence as it is is unclear and I suggest rephrasing, dividing or better explaining.

  Thank you for pointing at this error. We have corrected the sentence accordingly.

- L110: "*As it was noted above, the SVR model has two tunable parameters: cost and gamma...*" Actually, gamma is never mentioned. The authors mention "sigma" on L99. Please clarify.

  We apologize for this confusion. We meant the same parameter, 'gamma', which is sometimes referred in literature as 'sigma'. We now use term 'gamma' throughout the new version of the manuscript.

**3. Model validation**

- L160: Please cite  https://doi.org/10.1016/0022-1694(70)90255-6

  Thank you for suggesting the reference. We included a reference to Nash and Sutcliffe 1970 (line 273)

- L180: As mentioned in Comment 0.1, Mendoza and Western Pamir are not mentioned earlier in the text as data used for validation and are only introduced here.

  Introduction to Mendoza Andes and Western Pamir regions is now moved to a new section 'Data'  (lines 202-210).

- L199-200: Do the authors refer to Figure 4? If so, Figure 4 needs to be mentioned. See the comments about Figures.

  We appreciate this suggestion. We have introduced all figures in the text in the new version of the manuscript.

- L202: *"... the rain-to-snow transition modelled using the metadata of the 520 validation SNOTEL stations."*  Do the authors mean that there are observations/data on the transition between rain and snow for all the 520 stations? And how was that used in modelling? Please clarify.

  The main motivation behind this analysis is to have an understanding how the model simulates precipitation-snow partitioning during snow accumulation phase. The following new exert provide additional details in this regard (lines 255-262):

  > "Since the SNOTEL observations do not contain explicit information on precipitation-snow transition, we decided to use a sample of the dataset to simulate the transition depending on climate inputs (temperature variables) and topographical characteristics (e.g. elevation). More specifically we have filtered the SNOTEL observations that closely fall on this phase by selecting observations that meet the following non-exhaustive main criteria: 1) observations for October or November when precipitation is non-zero 2) average temperature (TAVG) is less than 10 or higher than -10°C, 3) accumulated SWE is less than 20mm. We then run the model using the obtained sample of observations and estimated solid fraction of precipitation simulated by the model, i.e. amount of dSWE estimated by the model in respect to precipitation amount."

- L206: *"... does not exceed 100%"* do the authors mean does not *reach* 100%?

  Yes, indeed, 'not reach 100%' is more appropriate here and we have rephrased this part accordingly.  Thank you for this correction.

- L210: I suggest justifying this sentence with a plot or a better explanation. Again, if this information is contained within some metadata, this needs to be explicitly stated.

Unfortunately, explicit rain-to-snow transition thresholds are not provided in the SNOTEL data. We assume that the updated description of $T_S$ and how it differs from the traditional temperature threshold (lines 274-280 and Figure 5) offers some explanation. Moreover, the default $T_S$ threshold exhibits satisfactory performance across the majority of validation stations, encompassing both SNOTEL and ESM-SnowMIP sites.

- L241: How did the authors calibrate Ts? Please clarify.

  We have included the following description into the text (lines 309-310):

  > "We calibrated $T_S$ for each of the stations with the objective of maximizing the Nash-Sutcliffe Efficiency of the model`s simulations with respect to observed SWE, and bounded the range of calibrated TS to -5 to +5 °C."

- L255-256: Can the authors verify this assumption? Shortly after, in the text, the authors write the same for the SnowMIP station SNB, so I assume it is possible?

  Thank you for these guiding questions. Here we made an assumption that simulations at larger margin of adjusted Ts likely led to overcalibration compensation, also implying a model limitation for locations susceptible to snow drifts. We now realize that beside the snow drifting, there might be several other contributing factors leading to overestimation of SWE, such as sublimation, effect of dense canopy, and rain-on-snow events. Unfortunately, we can not delineate/ verify these factors within the scope of this manuscript, but we believe they should be noted since they also define limitations of model. Therefore, we rewrote this passage in the following manner (lines 320-326):

  > "While the median of the adjusted TS values for all stations agrees with its default threshold (-1 °C), the density distribution also shows a high frequency of calibrated Ts resulting at the lowest bound of -5 °C (Figure 7f). This suggests that, in many cases where calibrated Ts values approach the lowest boundary, the model simulations might have been overcalibrated, resulting in error compensation. The overestimation of SWE at these locations can be attributed to several factors that the model does not account for, including effect of dense vegetation, wind induced snow-drift, sublimation, and rain-on-snow events which may be frequent phenomena in the mountain areas (Li et al., 2019; Boniface et al., 2015; Kirchner et al., 2014; Sexstone et al., 2018)."

  In addition, we have also noted these limitations in the "Model limitations" section.

- L292: The authors should explain the meaning of *"class balance accuracy"*.

  We have supplemented this sentence with a brief explanation of class balance accuracy and reference (lines 391-393):

  > "Overall pixel-wise accuracy of snow/no-snow detection for both regions was 92%, while the class-balanced accuracy, which takes into account the balance of class distribution (Branco et al., 2016), was 87% on average."

**4. Model sensitivity and uncertainty assessment**

- L305: Is there a reference for this method? If so, I suggest adding it.

  Yes, this method is explained in Fisher et al., 2018 and Greenwell et al., 2018. We have added these references into the sentence (line 409).

- L311: *"... depending on the phase considered ..."* Do the authors mean "precipitation phase"? Please clarify. Also, the reference is missing.

  We refer to two general phases of snow metamorphosis - snow accumulation and snow ablation. We edited the sentence in the revised version of the manuscript (lines 409-411):

  > "We applied the permutation-based feature importance analysis on the entire training dataset of the independent SNOTEL stations, as well as its subsamples representing snow accumulation or melt phases"

  Our apologies for the missing reference; it was supposed to be a cross-reference to the Figure 10 further down.

- L316: What do the authors mean by *"relative comparison"*? Please clarify.

  In the given context, "relative comparison" means that the importance of those topographic variables is made in relation to other variables used by the model. We rewrote this line in the text to make it clearer (lines 418-420):

  > "At first glance, the results suggest that topographic variables are among the least influential, but it should be noted that their significance is assessed in relation to other variables, some of which, such as precipitation and temperature, are more fundamental for accurate snowpack estimation (Günther et al., 2019)."

- L349: Please refer to Table 1 when addressing the different model settings.

  A cross-reference to the Table 1 has been included in the line 456

- L355: What do the authors mean by *"when outliers are controlled for"*? Please clarify.

  The boxplots in Figure 12 show extreme limits, which exclude outliers. More specifically, the minimum and maximum limits of the boxplots are determined by (1st Quartile - 1.5 * IQR) and (3rd Quartile + 1.5 * IQR), where IQR represents the interquartile range (Hu, 2020). To prevent confusion, we have removed the phrase 'when outliers are controlled for' from the sentence.

**5. Summary**

- L375: The concept of equifinality is only addressed at the end of the paper but it is never mentioned earlier. The most important papers on equifinality are not cited (see

https://doi.org/10.1016/0022-1694(89)90101-7, https://doi.org/10.1016/0309-1708(93)90028-E, https://doi.org/10.1016/j.jhydrol.2005.07.007). If overcoming equifinality is one of the aims of the paper, this needs to be addressed in the Introduction and also in the discussion of the results. And additionally, how does the model improve equifinality? This needs to be explained and justified. The results shown in Figure 12, for example, seem contradictory to this sentence, because there the authors show that one can obtain similarly good model performances with different sets of parameters.

Thank you for suggested references. In this sentence we rather refer to the challenge of calibrating multiple parameters in hydrological and snow modelling. We have briefly introduced issue of equifinality in the introduction (lines 39-42):

> "The two types of snow models usually require adjustment of internal parameters that characterize embedded processes. Depending on the complexity of a model, calibrating its parameters can become a computational burden and introduce challenges related to the equifinality of model parameters (Beven, 1993, 2006; Günther et al., 2020)."

We have also corrected and expanded respective exert in the "Summary and Conclusions" section with the following (lines 537-543):

> "In addition to avoiding computationally demanding calibration, GEMS may also alleviate the challenge of equifinality of model parameters that is pertinent to modelling environmental systems. The issue of equifinality is particularly pronounced in hydrological modelling (), where even relatively simple snow modules require the calibration of at least two parameters: the precipitation-snow threshold and the degree-melt coefficient. Considering that there are many other parameters for remaining components of a hydrological model, it would be easy to end up with multiple combinations of optimal parameters. In contrast GEMS model shows generally plausible performance in diverse climatic and topographic conditions upon using the default value of TS. "

Figure 12 shows performance of four GEMS models that differ in a number of required inputs, but contain only a single parameter ($T_S$) which can be adjusted. All four models' performances depicted in figure 12 were obtained by using the default value of the $T_S$ (-1°C)

L383-385: This sentence is not clear. What do the authors mean by *"instrumental"*?

We edited the sentence (line 525), by replacing '*instrumental*' with '*helpful*'. Here we meant that "*balance (in) complexity, data requirement, and transferability... could be helpful for operational monitoring and hydrological modelling in data scarce domains.*"

- L385: Similarly for the equifinality, the problem of finding empirical relations and parametrizations is never addressed before in the text. If this is one of the aims of the paper, it needs to be addressed in the Introduction accompanied by proper references (as parametrizations of different kinds are already widely used in snow/hydrological modelling).

Thank you for raising this. We now recognize that the statement in this sentence may have been too assertive and requires further verification. We have removed this sentence from the manuscript.

- Please consider mentioning the undercatch selection issue within the Model limitation section.

By filtering observations for precipitation undercatch, we assume that the evaluation dataset is comparatively free of this issue. However our selection algorithm also filtered records where inconsistencies between accumulated precipitation and SWE may be reasoned by wind-induced snow-drift. Disentangling these two phenomena is challenging without further research. The model cannot capture/simulate snow-drifts, we acknowledge this limitation in lines 441-443 and explicitly stated it in lines 498-500.

---          *FIGURES*          ---

**General comments:**

- When a figure is composed by different subplots, as it is often the case in this paper, something that enhances clarity very much is naming each subplot differently, for example with letters like (a), (b)... And then, throughout the text, referring to each subplot like Figure 5a, Figure 5b etc.

- I suggest improving the figure referencing generally and throughout the whole text: often the authors describe the results referring to specific subplots of a same Figure by only mentioning the general Figure once at the beginning of the paragraph. Referring to each specific subplot before introducing each finding highlighted by the subplot increases clarity significantly.

Thank you for these recommendations. We have reorganized the figures accordingly, and ensured they are properly introduced and referenced in the text.

**Specific comments:**

- Figure 2: Axes ticks and labels (latitude, longitude) are missing, legend is missing.

- Figure 3: Axes labels are missing.

- Figure 6: Left plots: missing adimensional symbol for NSE ([-]), missing unit for snow meltout date error (days?), missing y-axis label. Right plots: Missing axes ticks and labels (latitude, longitude).

- Figure 7: Same as above.

- Figure 8: y-axis label and units are missing.

- Figure 11: "Latitude" is spelled wrong, missing units, missing y-axis ticks and labels.

Thank you for pointing out at these deficiencies. We have corrected these figures accordingly.

[revised manuscript text omitted]